# Arabidopsis telomerase takes off by uncoupling enzyme activity from telomere length maintenance in space

Borja Barbero Barcenilla[1], Alexander D. Meyers[2,3,4,8], Claudia Castillo-González [1,8], Pierce Young[1], Ji-Hee Min[1], Jiarui Song [1], Chinmay Phadke[1], Eric Land[5], Emma Canaday[2,3], Imara Y. Perera [5], Susan M. Bailey[6], Roberto Aquilano[7], Sarah E. Wyatt [2,3] ✉ & Dorothy E. Shippen [1] ✉

Spaceflight-induced changes in astronaut telomeres have garnered significant attention in recent years. While plants represent an essential component of future long-duration space travel, the impacts of spaceflight on plant telomeres and telomerase have not been examined. Here we report on the telomere dynamics of *Arabidopsis thaliana* grown aboard the International Space Station. We observe no changes in telomere length in space-flown Arabidopsis seedlings, despite a dramatic increase in telomerase activity (up to 150-fold in roots), as well as elevated genome oxidation. Ground-based follow up studies provide further evidence that telomerase is induced by different environmental stressors, but its activity is uncoupled from telomere length. Supporting this conclusion, genetically engineered super-telomerase lines with enhanced telomerase activity maintain wildtype telomere length. Finally, genome oxidation is inversely correlated with telomerase activity levels. We propose a redox protective capacity for Arabidopsis telomerase that may promote survivability in harsh environments.

Eukaryotic chromosomes are capped with telomeres, non-coding G-rich DNA repeats and associated proteins that promote genome stability. Telomeric DNA tracts are highly dynamic, but in each species an optimal length setpoint is established[1]. The predominant telomere length maintenance pathway utilizes telomerase, an enzyme whose expression is largely confined to proliferating cell populations. Telomere length is influenced by a multitude of genetic, epigenetic and environmental factors[2,3] with both critically shortened and hyper-elongated telomeres being detrimental in humans[4,5]. *A. thaliana* mutants with short telomeres exhibit altered flowering time and

reduced fitness[6,7], while extreme shortening below a 1 kb size threshold triggers end-to-end chromosome fusions and massive genome instability[7,8]. Arabidopsis mutants with grossly extended telomeres undergo stochastic telomere resection[9] and this propensity is increased in dry environments[6], raising the possibility that ultra-long telomeres (at least in a Ku defective background) may be more fragile in response to stress. Interestingly, natural accessions with longer telomeres accumulate more telomeric 8-oxoG, but have less chromosomal oxidation than natural accessions with short telomeres[10]. Telomeres are thus postulated to serve as sentries of physiological and

[1]Department of Biochemistry and Biophysics, Texas A&M University, 2128 TAMU, College Station, TX 77843, USA. [2]Department of Environmental and Plant Biology, Ohio University, Athens, OH 45701, USA. [3]Molecular and Cellular Biology Program, Ohio University, Athens, OH 45701, USA. [4]NASA Postdoctoral Program, Oak Ridge Associated Universities, Kennedy Space Center FL, Merritt Island, FL 32899, USA. [5]Department of Plant and Microbial Biology, North Carolina State University, Raleigh, NC 27695, USA. [6]Department of Environmental and Radiological Health Sciences, Colorado State University, Fort Collins, CO 80523, USA. [7]National Technological University, Rosario Regional Faculty, Zeballos 1341, S2000 Rosario, Argentina. [8]These authors contributed equally: Alexander D. Meyers, Claudia Castillo-González. ✉e-mail: wyatts@ohio.edu; dorothy.shippen@ag.tamu.edu

environmental stressors as well as biomarkers for proliferative capacity and lifespan in a variety of different species[2,11–14].

The extreme environmental factors that accompany spaceflight include microgravity and space radiation exposure. Plant experiments conducted during spaceflight show that proliferating cells endure DNA damage, decoupling of cell proliferation from cell growth, cell cycle alterations, impaired ribosome biogenesis, and modification of the epigenome[15–18]. Radiation exposure, in particular, causes a range of stress responses from stimulatory effects at low doses (enhanced germination and seed growth), to deleterious repercussions (delayed development) at intermediate levels, and pronounced negative effects (plant death) at high doses[19–23]. The severity of such effects differs among species, cultivars, and with plant age, physiology, morphology, and genome structure[24]. Mechanisms underlying spaceflight-induced plant stress responses, genomic adaptation to spaceflight environments, and long-term viability under different space radiation scenarios remain poorly understood.

Telomere length homeostasis and genome stability were recently evaluated in twin astronauts Scott and Mark Kelly, along with a cohort of unrelated astronauts, before, during, and after 1-year or shorter missions aboard the ISS[25–27], and most recently in the Inspiration4 civilian crew during their 3-day mission[28]. Telomeres were significantly elongated during spaceflight in all crewmembers and in all in-flight samples analyzed (blood, and in one case also urine), irrespective of mission duration or means of measurement. In addition, mitochondrial dysregulation, increased oxidative stress and inflammation (evidenced by elevated 8-oxoG, TNF, PGF2) as well as genome instability were observed during spaceflight in these crew members, and astronauts in general[25,27,29–31]. Consistent with a rapid human response to spaceflight, telomere elongation and upregulation of pathways related to oxidative stress were evident during and post the 3-day high elevation orbital mission for all four Inspiration4 crew members[28].

Elevated telomerase activity during spaceflight could account for the observed telomere elongation, but the unavoidable transit time and temperature conditions during ambient sample return to Earth precluded direct assessment of telomerase activity in astronaut samples[25,26]. Strikingly, telomere tracts shortened rapidly upon return to Earth, and although individual differences were observed, most crew members had many more short telomeres after spaceflight than they had before[25–27]. Spaceflight also appears to alter telomere dynamics in *Caenorhabditis elegans*, which experienced net telomere elongation during an 11 day mission onboard the ISS[32]. Increased understanding of telomere length dynamics and regulation during space travel may provide insight into the potential short and long-term health implications for biological systems in general, an important consideration for future exploration missions.

Here we examine the status of telomeres, telomerase and genome oxidation in *A. thaliana* seedlings grown aboard ISS. We report increased genome oxidation and a strong induction of telomerase enzyme activity in the roots of space-grown plants that is not correlated with an increase in telomere length. Notably, ground experiments show that genetically engineered plants that grossly overexpress telomerase also fail to extend their telomeres beyond wildtype length, but genome oxidation is substantially reduced. These findings argue that telomerase activity and telomere length homeostasis are uncoupled in *A. thaliana*, and point to a non-canonical role for telomerase in redox biology that may help enhance the survivability of plants during long-duration space missions.

## Results and discussion
### Plant telomere length remains unchanged during spaceflight
We examined the impact of spaceflight on telomere dynamics in 12-day-old *A. thaliana* seedlings grown in Veggie hardware[33,34] aboard the ISS. Unused plates available from the APEx-07 spaceflight were employed for our study (Supplementary Fig. 1a–c). The primary goal of the APEx-07 was to examine post-transcriptional regulation of gene expression using the *A. thaliana 35S::HF-RPL18* line (*35S::HF-RPL18*, epitope tagged, large ribosomal subunit[35] for sequencing ribosome-associated RNAs, i.e. Translating Ribosome Affinity Purification (TRAP)-seq[33]). Ground controls were established at the Kennedy Space Center under conditions similar to those of APEx-07 spaceflight. For the current study, we separately analyzed DNA and protein from roots and shoots of space-grown plants, plants grown in a Random Positioning Machine (RPM), and in 1*g* ground controls, because these organs display different susceptibility to spaceflight conditions[36]. For root samples, each biological replicate consisted of a pool of 5–6 roots. For shoot samples, biological replicates comprised 3–4 shoots.

We employed several strategies to assess telomere length in space-flown plants. First, we compared bulk telomere length in shoots and roots using Terminal Restriction Fragment (TRF) analysis (Fig. 1a, b and Supplementary Fig. 2a). We found no statistically significant difference in mean telomere length in either organ relative to the controls. In shoots, mean telomere length in ground controls was ~3.1 kb ($n = 3$) and ~2.9 kb ($n = 3$) in spaceflight ($p = 0.5032$ by unpaired two-tailed Welch's *t*-test) (Fig. 1a). In roots, mean telomere length in ground controls was ~2.78 kb ($n = 3$) and ~2.75 kb ($n = 3$) in spaceflight ($p = 0.8684$ by unpaired two-tailed Welch's *t*-test). We also examined telomere length on individual chromosome arms using Primer Extension Telomere Repeat Amplification (PETRA)[8] (Fig. 1c, d). This method amplifies telomeric DNA using primers directed at unique subtelomeric sequences adjacent to each telomere repeat array. PETRA was conducted for four chromosome arms in each biological replicate, and images were quantified with WALTER[37] to determine mean telomere length (Fig. 1e, f). Combined PETRA and TRF data indicate that telomere length is very similar in roots and shoots (Supplementary Fig. 2), consistent with prior studies showing telomere length is remarkably stable during *A. thaliana* development[38]. PETRA also revealed no statistically significant difference in telomere length in space-grown plants versus ground controls, regardless of whether we compared individual chromosome arms (two-way RM ANOVA) (Fig. 1e in shoots, 1f in roots) or the average telomere length of all measured chromosomes in each individual. Average telomere length ($n = 8$, four chromosome arms) from the shoots of ground control samples was 3645 bp, while flight shoot telomeres averaged 3537 bp ($n = 8$, four chromosome arms), for a net difference of −107 bp ($p = 0.2865$ by unpaired two-tailed Welch's *t*-test, $n_{control} = 32$, $n_{spaceflight} = 32$) (Fig. 1c, e and Supplementary Fig. 2b, d). There was also no significant change in flight root telomere length (average of 3608 bp, $n = 6$, four chromosome arms) relative to ground controls (3596 bp, $n = 6$, four chromosome arms) for a net difference decrease of 18 bp ($p = 0.8425$ by unpaired two-tailed Welch's *t*-test, $n_{control} = 23$, $n_{spaceflight} = 24$) (Fig. 1d, f and Supplementary Fig. 2c, d). Finally, we corroborated these results by a quantitative PCR-based telomere content assay[10,39] defined as the ratio of telomere to single copy gene content (T/S). Average T/S in shoots from ISS samples (0.66, $n = 3$) and ground controls (0.67, $n = 3$) were not statistically different ($p = 0.0566$ by unpaired two-tailed Welch's *t*-test) (Supplementary Fig. 2e). Likewise, there was no difference in the average T/S for roots in space (0.65, $n = 5$) and ground controls (0.66, $n = 6$) ($p = 0.4089$ by unpaired two-tailed Welch's *t*-test; Supplementary Fig. 2e). These findings indicate that telomere length homeostasis is maintained for *A. thaliana* space-grown plants.

### Spaceflight leads to a dramatic increase in telomerase activity in Arabidopsis roots
We next assessed telomerase enzyme activity using the gel-based Telomere Repeat Amplification Protocol (TRAP) and the quantitative version of this assay (Q-TRAP)[40] (Fig. 2). TRAP (Fig. 2a) and Q-TRAP (Fig. 2b, c) both showed elevated telomerase activity in space-flown shoots and roots compared to ground controls. Telomerase was an

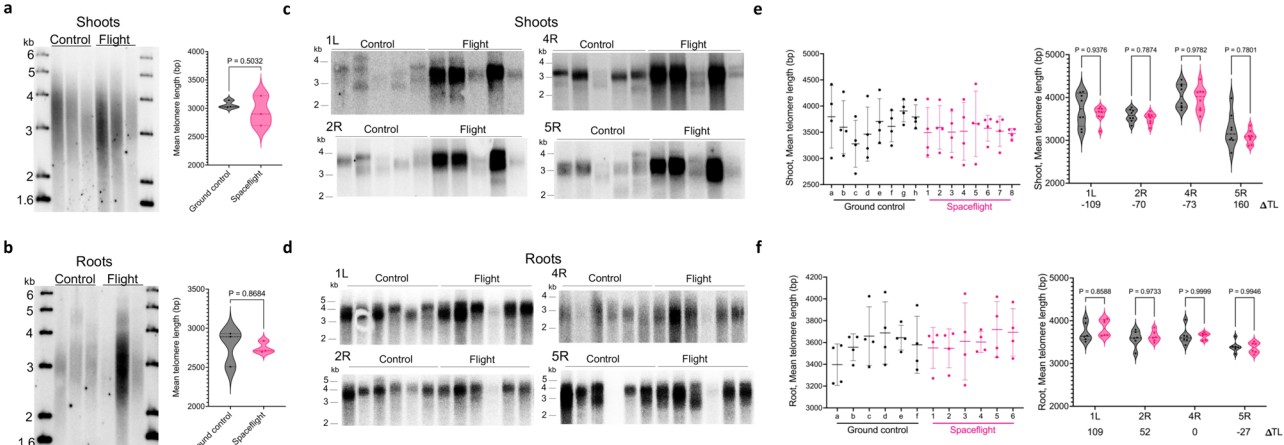

**Fig. 1 | Telomere length homeostasis is unperturbed in space-grown Arabidopsis.** Telomere length analysis of shoots (**a**, **c**, **e**) and roots (**b**, **d**, **f**) from 12-day-old *A. thaliana* seedlings grown aboard the ISS. **a**, **b** (left) Results from TRF analysis conducted with shoots (**a**) or roots (**b**). Biological replicates for space-flown shoots or roots ($n = 3$) and corresponding ground controls (grown under similar conditions at Kennedy Space Center) ($n = 3$). Each lane represents a single biological replicate. (right) Mean telomere length determined from TRF data (on left) analyzed by WALTER. Data displayed as violin plots. In shoots, mean telomere length for ground controls was ~3.1 kb and ~2.9 kb for spaceflight ($p = 0.5032$ by unpaired two-tailed Welch's *t*-test). In roots, mean telomere length for ground controls was ~2.78 kb and ~2.75 kb for spaceflight ($p = 0.8684$ by unpaired two-tailed Welch's *t*-test). **c**, **d** PETRA results for space-flown shoots (**c**) and space-flown roots (**d**) with their corresponding ground controls. Biological replicates for shoots $n = 5$ for both flight and ground controls, and for roots $n = 6$ for both flight and ground controls. Results are shown for the left arm of chromosome 1 (1L), the right arm of chromosome 2 (2R), the right arm of chromosome 4 (4R), and the right arm of chromosome 5 (5R). PETRA products for several space-flown samples are more

abundant relative to the ground controls. The reason is unknown. **e** (left) PETRA quantification of the mean telomere length for all the chromosome arms analyzed from each biological shoot sample determined using WALTER ($n = 8$). Data for three shoot biological replicates shown in Supplemental Fig. 2c were included. Results represent mean with SD. (right) Combined mean telomere length for each chromosome arm measured by PETRA from shoots analyzed by WALTER with relative telomere length changes in base pairs (ΔTL) indicated ($n = 8$). Data for three shoot biological replicates shown in Supplemental Fig. 2c were included. Data shown as violin plots. *P* calculated by two-way RM ANOVA. **f** (left) PETRA quantification of the mean telomere length for all the chromosome arms analyzed from each biological root sample determined by WALTER ($n = 6$; from data in Fig. 1d). Results represent mean with SD. (right) Combined mean telomere length for each chromosome arm measured by PETRA from roots analyzed by WALTER with relative telomere length changes in base pairs (ΔTL). ($n = 6$; from data in Fig. 1d). Data shown as violin plots with *p* calculated by two-way RM ANOVA. Ground control data in gray and spaceflight in pink. Source data provided as Source Data File.

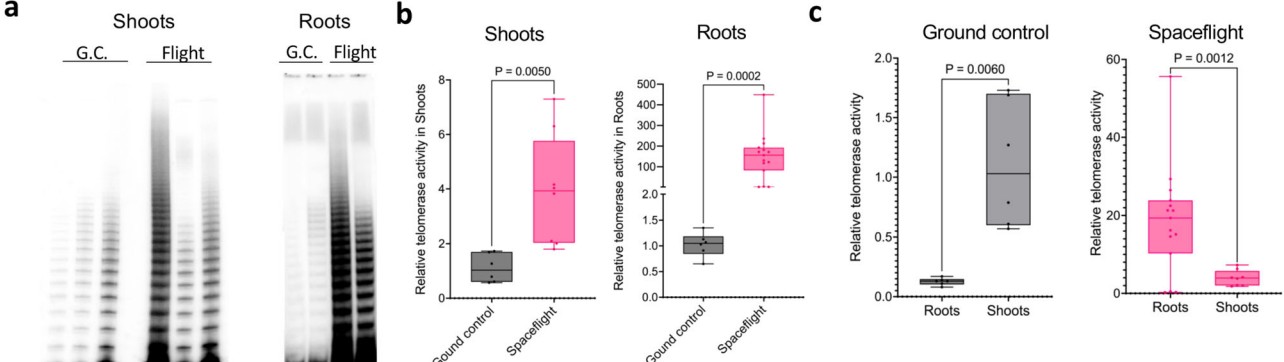

**Fig. 2 | Increased telomerase activity in roots and shoots of space-flown *A thaliana*. a** TRAP performed with 12-day-old seedlings grown aboard the ISS and their corresponding ground controls (G.C.). For shoots (left) $n = 3$, and roots (right) $n = 2$. **b** Results of Q-TRAP assays for shoots (left) and roots (right) from 12-day-old flight seedlings and ground controls. Biological samples are distinct from those analyzed in Fig. 2a. For analysis of telomerase in shoots, biological replicates of ground controls $n = 6$ and flight samples $n = 8$. *P* calculated by unpaired two-tailed Welch's *t*-test, $n_{control} = 6$, $n_{spaceflight} = 8$, $p = 0.005$. For analysis of telomerase in roots, biological replicates of ground controls $n = 6$ and flight samples $n = 15$. *P* calculated by unpaired two-tailed Welch's *t*-test, $n_{control} = 6$, $n_{spaceflight} = 15$,

$p = 0.0002$. Three technical replicates were performed for each biological replicate. Within the boxplot, the middle line represents the median, box boundaries signify 25th and 75th percentiles, and whiskers the lowest and highest values. **c** Relative telomerase activity measured by Q-TRAP (from Fig. 2b) plotted for ground control and space-flown roots and shoots. *P* calculated by unpaired two-tailed Welch's *t*-test, $p = 0.006$ and $p = 0.0012$, respectively. Box and whisker plots show minimum to maximum values and all biological points. Within the boxplot, the middle line represents the median, box boundaries signify 25th and 75th percentiles, and whiskers the lowest and highest values. Source data provided as Source Data File.

average of ~4-fold higher in space-flown shoots (unpaired two-tailed Welch's *t*-test, $n_{control} = 6$, $n_{spaceflight} = 8$, $p = 0.005$), and strikingly, approximately 150-fold higher in roots (Fig. 2b) (unpaired two-tailed Welch's *t*-test, $n_{control} = 6$, $n_{spaceflight} = 15$, $p = 0.0002$). Although the baseline level of telomerase activity in root samples ($n = 6$) was ~0.13

times that of the average shoot (unpaired two-tailed Welch's *t*-test, $n_{roots} = 6$, $n_{shoots} = 6$, $p = 0.006$), the enzyme activity in space-flown roots versus space-flown shoots was still substantially higher (unpaired two-tailed Welch's *t*-test, $n_{roots} = 15$, $n_{shoots} = 8$, $p = 0.0012$; Fig. 2c). Under normal conditions expression of telomerase reverse

transcriptase (TERT) is restricted to meristematic cells[41,42] (single-cell transcriptomics data also analyzed from ref. [43]). Other studies indicate the *AtTERT* promoter is regulated by multiple methylation pathways and *AtTERT* expression is induced when the plant is exposed to high-salt media, a known inducer of oxidative stress[44]. Whether stress increases meristematic TERT or expands its cellular niches remains to be studied and might imply a much higher local telomerase activity than what can be measured here. Notably, qRT-PCR of isolated roots and shoots of space-flown seedlings (Supplementary Fig. 3) and re-analysis of spaceflight RNAseq data[45–48] showed no change in expression levels of *AtTERT, AtTR*, and several other telomerase-associated genes. Thus, the mechanism of telomerase induction in response to spaceflight is unknown.

### Increased telomerase activity does not alter telomere length in Arabidopsis

Additional on-ground investigation indicated that telomerase activity is elevated by treatments that result in oxidative stress including exposure to light stress[10,49–51] (Fig. 3). We subjected plants to two weeks of either a control light regime (12 h) or constant (24 h) light. Plants exposed to constant light showed increased $H_2O_2$ concentration from

an average of ~5.1 nmol/g of dry weight tissue ($n = 4$) in control conditions to ~8.8 nmol/g ($n = 3$) in plants subjected to constant light ($p = 0.0015$ by unpaired, two-tailed Welch's $t$-test; Fig. 3a) and a 50-fold higher accumulation of *Rd29b* (AT5G52300) transcripts, a marker for a variety of abiotic stresses[52] compared to control samples ($p = 0.0006$ by two-way ANOVA test, $n_{control} = 3$ and $n_{lightstress} = 3$; Fig. 3b).

Q-TRAP was then performed on flower bundles (open and uno-pened floral buds) as these organs bear reproductive shoot apical meristems, a natural niche of telomerase expression. Telomerase activity increased by ~12 fold upon light-induced oxidative stress relative to controls (unpaired two-tailed Welch's $t$-test, $n_{control} = 4$, $n_{lightstress} = 4$, $p < 0.0001$; Fig. 3c). As with plants grown aboard the ISS, light stress did not trigger a significant change in gene expression of *AtTERT* ($p > 0.9999$ by two-way ANOVA, $n_{control} = 3$ and $n_{lightstress} = 3$; Fig. 3b). We also saw no significant change in telomere length in light stress treated plants when we compared individual telomeres (two-way RM ANOVA, $n = 3$, three chromosome arms; Fig. 3d) or the average telomere length of all measured chromosome ends in each individual ($p = 0.2548$ by unpaired two-tailed Welch's $t$-test, $n_{control} = 8$, $n_{lightstress} = 9$; Fig. 3e–g). These results indicate that telomerase induction is not unique to spaceflight, and are consistent with prior studies

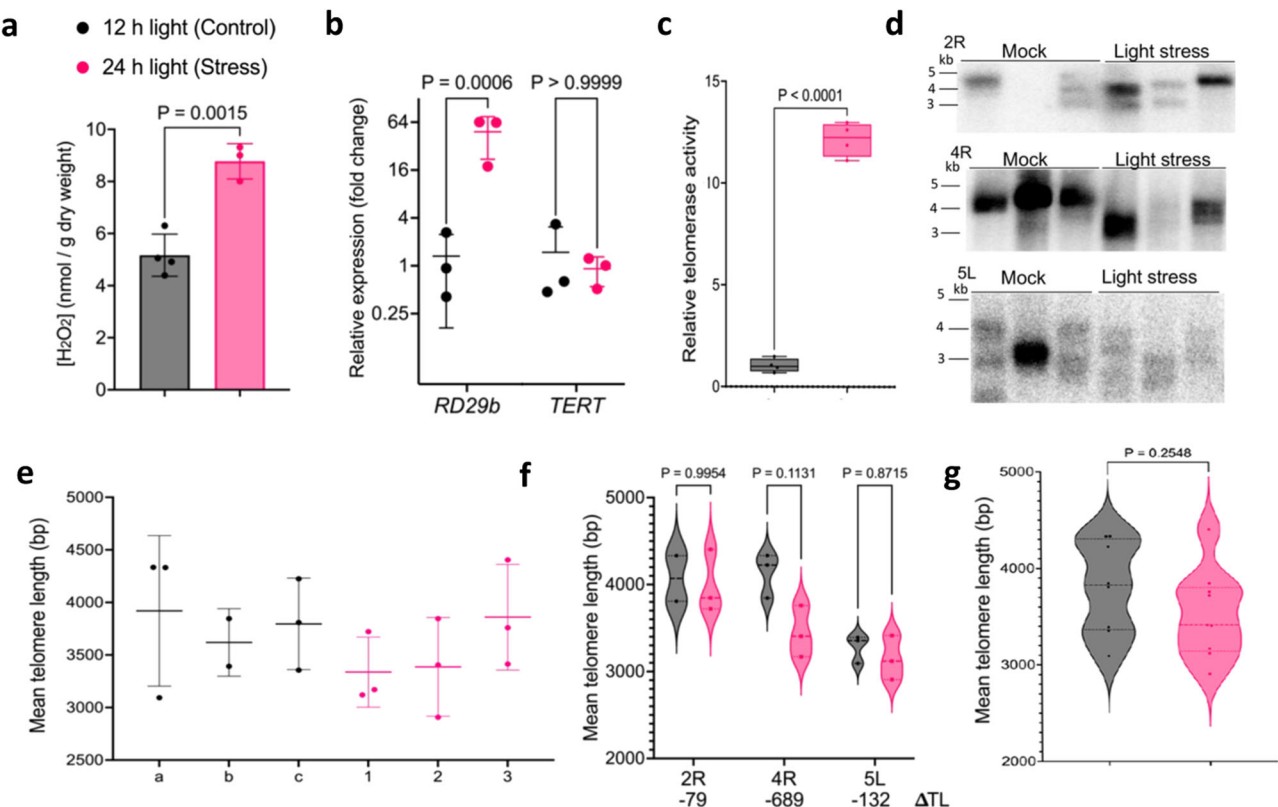

**Fig. 3 | Telomere length and telomerase activity analysis of plants subjected to light stress.** 4-week-old *A. thaliana* were subjected to two weeks of either a control light regime (12 h) or constant (24 h) light to induce oxidative stress. **a** Hydrogen peroxide measured in rosette leaves using potassium iodide. Results represent mean with *SD*. Control ($n = 4$) and light stress ($n = 3$) with two technical replicates per biological sample. $P = 0.0015$ by unpaired, two-tailed Welch's $t$-test. **b** Relative transcript levels measured by qRT-PCR of control and light stress samples of *RD29b* and *TERT*. Results represent mean with SD with two technical replicates performed per biological sample. $P = 0.0006$ by two-way ANOVA, $n_{control} = 3$ and $n_{lightstress} = 3$ for *RD29b* and $p > 0.9999$ by two-way ANOVA, $n_{control} = 3$ and $n_{lightstress} = 3$ for TERT. **c** Relative telomerase activity in 6-week-old flower bundles measured by Q-TRAP. For both control and light stress samples ($n = 4$). For every biological sample, three technical repeats were performed. Within the boxplot, the middle line represents the median, box boundaries signify 25th and 75th percentiles, and whiskers the

lowest and highest values. Statistical analysis performed by unpaired two-tailed Welch's $t$-test, $p < 0.0001$. **d** PETRA results for leaves of control ($n = 3$) and light stress samples ($n = 3$) for chromosome arms 2R, 4R, and 5L are shown. Each lane represents a biological replicate, DNA was obtained from rossette leaves. **e** Telomere length of each biological sample in panel **d**, shown as mean with SD of the combined telomere lengths of all chromosome arms analyzed by PETRA and by WALTER ($n_{control} = 8$, $n_{lightstress} = 9$). **f** Combined mean telomere length of individual chromosome arms for each biological sample measured by PETRA and analyzed by WALTER. P obtained by two-way RM ANOVA, $n = 3$, three chromosome arms. **g** Combined mean telomere length of all chromosome arms for all biological sample measured by PETRA and analyzed by WALTER. $P = 0.2548$ by unpaired two-tailed Welch's $t$-test, $n_{control} = 8$, $n_{lightstress} = 9$. Control data in gray and light stress data in pink. Source data provided as Source Data File.

showing that *A. thaliana* sustains telomere length homeostasis in the face of different environmental stressors including heat, drought and flood conditions[6,53].

Since we do not know the extent to which telomerase is induced in space-flown plants (increased expression in a few meristematic cells or expanded expression in other niches), it is possible that telomeres are elongated in a subpopulation of cells which would preclude detection by standard methods. To further investigate the apparent uncoupling of telomerase induction and telomere elongation, we artificially raised telomerase activity levels in transgenic plants. Core telomerase subunits *AtTERT* and *AtTR* were expressed using the 35S and U6 snRNA promoters, respectively, in a *tert* null background to create super-telomerase plants. In parallel, we produced complementation-telomerase plants in this same background that expressed *AtTERT* from its native promoter and *AtTR* from the U6 snRNA promoter. In the second generation (T2) transformants, Q-TRAP performed on two independent complementation-telomerase lines, 8-3-2 and 7-4-2, showed 1x and 2x the wildtype telomerase activity, respectively. By comparison, two independent super-telomerase lines, 5-2-1 and 10-3-3, displayed 31x and 135x telomerase activity, respectively (Supplementary Fig. 4). For the complementation-telomerase lines, TRF analysis in the third generation (T3) revealed telomeres that were at or slightly above wildtype range of 2–5 kb[54] (Fig. 4a–c), indicating that the *tert* mutation was complemented.

In contrast, although telomeres were extended in super-telomerase plants compared to *tert* mutants, they were not elongated above the wildtype size range in T3. For the 5-2-1 line, the average length of telomeres was 2073 bp compared to the Col-0 control at 2890 bp (Brown-Forsythe ANOVA, $n = 3$, $p = 0.0027$). For the 10-3-3 line, the average length was higher than in the other super-telomerase line (2617 bp) (Brown-Forsythe ANOVA, $n = 3$, $p = 0.9979$). However, telomeres were much shorter for two of the 10-3-3 individuals (average of 2272 bp) and sharp bands in their TRF profiles were observed (Fig. 4a, 10−3−3D and 10−3−3E), indicative of ineffective telomerase engagement and/or activity at telomeres[7]. We evaluated the T4 progeny of the 5-2-1 line and TRFs of four T4 individuals revealed no significant change in telomere length from T3 to T4 (Brown-Forsythe ANOVA, $p = 0.1552$; Fig. 4b, c). Q-TRAP analysis of T4 individuals 5-2-1-C and 10-3-3-F showed super-telomerase activities of ~17 and ~31-fold, respectively, compared to wildtype plants (one-way ANOVA, $n_{Col-0} = 6$, $n_{5-2-1-C} = 5$, $n_{10-3-3-F} = 5$, $p_{5-2-1-C} = 0.0279$ and $p_{10-3-3-F} = 0.0003$; Fig. 4d). These findings agree with previous studies on the complementation of AtTERT function[55–57]. Notably, telomerase activity in both 5-2-1 and 10-3-3 transgenic lineages decreased from T3 to T4 (Fig. 4d and Supplementary Fig. 4), suggesting that persistently high levels of telomerase may not be well-tolerated.

One caveat with 35S promoter expression is that it is largely confined to mature, differentiated tissues[58], and hence *35S:TERT* may not be effective in extending telomeres outside of meristematic tissues where the enzyme is naturally expressed. This notion is refuted by a recent study with a 35S-driven hypomorphic *AtTERT* allele[57]. Telomerase activity levels are well below wildtype in this line and plants have very short telomeres. Nevertheless, if the blunt-end telomere capping complex Ku is mutated in this background, there is immediate (first plant generation) and dramatic elongation of bulk telomeres by telomerase[57,59]. Because bulk telomeres can be grossly extended, even when telomerase levels are much lower than in wildtype, these findings argue that *35S:TERT* expression is not limiting for telomere extension. An alternative explanation is that chromosome ends must be made accessible for telomerase-mediated telomere elongation. In support of this model are data from Arabidopsis mutants with constitutive telomerase expression in leaves, a setting where telomerase is not normally expressed. Telomeres are not extended beyond the wildtype length in these leaves[60]. Finally, we note that the telomeres of budding yeast are recalcitrant to telomerase-mediated extension unless chromosome ends are in an extendable chromatin conformation[61,62].

Indeed, a complex interplay between telomerase and telomere-associated proteins, DNA replication and repair machinery, and the telomere epigenome establishes telomere length homeostasis[63]. A similar mechanism may operate in Arabidopsis.

## Spaceflight triggers increased genomic 8-oxoG and altered organellar homeostasis in Arabidopsis

Omics studies demonstrate that spaceflight is associated with significant increases in oxidative stress in plants[45,64–68]. We directly assessed genomic DNA oxidation in shoots and roots of space-flown seedlings using a commercially available ELISA assay kit to quantify 8-oxoG[10], the most common oxidative lesion in DNA[69]. We found a ~1.5-fold increase in 8-oxoG of shoots relative to ground controls (unpaired two-tailed Welch's *t*-test, $n_{control} = 6$, $n_{spaceflight} = 10$, $p = 0.0225$; Fig. 5a, left). In agreement with previous studies showing that the root apical meristem is highly sensitive to ROS[70], 8-oxoG levels were also higher in the roots of space-grown plants with a ~1.6-fold increase relative to ground controls (unpaired two-tailed Welch's *t*-test, $n_{control} = 12$, $n_{spaceflight} = 10$, $p = 0.0041$; Fig. 5a, right). Thus, we detected an increase in 8-oxoG lesions of over 50% upon germination during spaceflight in Arabidopsis.

Oxidative stress can also manifest as alterations in organellar morphology and increased mitochondria, chloroplasts, vacuoles, and amyloplasts[71,72]. We measured the abundance of mtDNA and cpDNA in space-flown plants as a proxy for organellar function. DNA from pooled shoots was subjected to qPCR to amplify mitochondria-encoded genes *nad6 (ATMG00270)*, *cox1* (ATMG01360), *atp1* (ATMG01190) and *rps4 (ATMG00290)*, and chloroplast-encoded genes *clpP (ATCG00670)*, *psbA (ATCG00020)* and *ndhH (ATCG01110)* along with *RpoTp* (AT2G24120) and *RpoTm* (AT1G68990) as nuclear controls[73]. Consistent with altered redox homeostasis[74], space-grown seedlings exhibited a 1.7-fold increase in mtDNA relative to ground controls, and a 2.5-fold increase in cpDNA (unpaired two-tailed Welch's test, 17 total combined measurements from ground control and 19 total combined measurements from spaceflight, from 7 and 8 biological samples of each treatment, respectively, $p_{mitochondria} = 0.0024$, $p_{chloroplast} = 0.0013$; Fig. 5b and Supplementary Fig. 5).

## Super-telomerase mutants display reduced genomic oxidation

Mammalian studies highlight an intriguing, but enigmatic connection between telomerase and redox homeostasis, especially with respect to mitochondrial function[75–77]. Because we observed greater accumulation of organellar DNA in plants germinated aboard the ISS as well as increased telomerase activity, we investigated whether these phenotypes were related. We evaluated mt and cpDNA content, relative to nuclear DNA, in shoots and roots of 7-day-old seedlings of Col-0 wildtype, second generation (G2) *tert* mutants and super-telomerase plants. Under normal conditions, we did not detect statistically significant changes in organellar DNA abundance, regardless of organelle, tissue, or genotype (6 biological replicates, 18 measurements per organelle, one-way ANOVA, $p > 0.9999$; Fig. 5c).

We also tested if changes in organellar DNA abundance would be evident in stress conditions by treating plants with radiomimetic drug zeocin. Stress induction was confirmed by qRT-PCR of the DNA repair gene, *BRCA1* (AT4G21070), which was induced ~200-fold ($n = 3$) upon zeocin treatment (Unpaired, two-tailed Welch's *t*-test, $p = 0.0131$) (Supplementary Fig. 6a). As in space and with light stress, telomerase activity was significantly induced by zeocin treatment (~10-fold, $n_{mock} = 5$, $n_{zeocin} = 6$, by two-tailed Mann-Whitney test, $p = 0.0043$; Supplementary Fig. 6b), without detectable transcriptional activation of *AtTERT* as tested by qRT-PCR (unpaired two-tailed Welch's *t*-test, $n = 3$, $p = 0.1867$; Supplementary Fig. 6c). Quantification of organellar DNA failed to reveal any changes, regardless of organelle, tissue or genotype (6 biological replicates, 18 measurements per organelle, one-way ANOVA; Fig. 5c and Supplementary Fig. 6d, e). Thus, our data do not support an

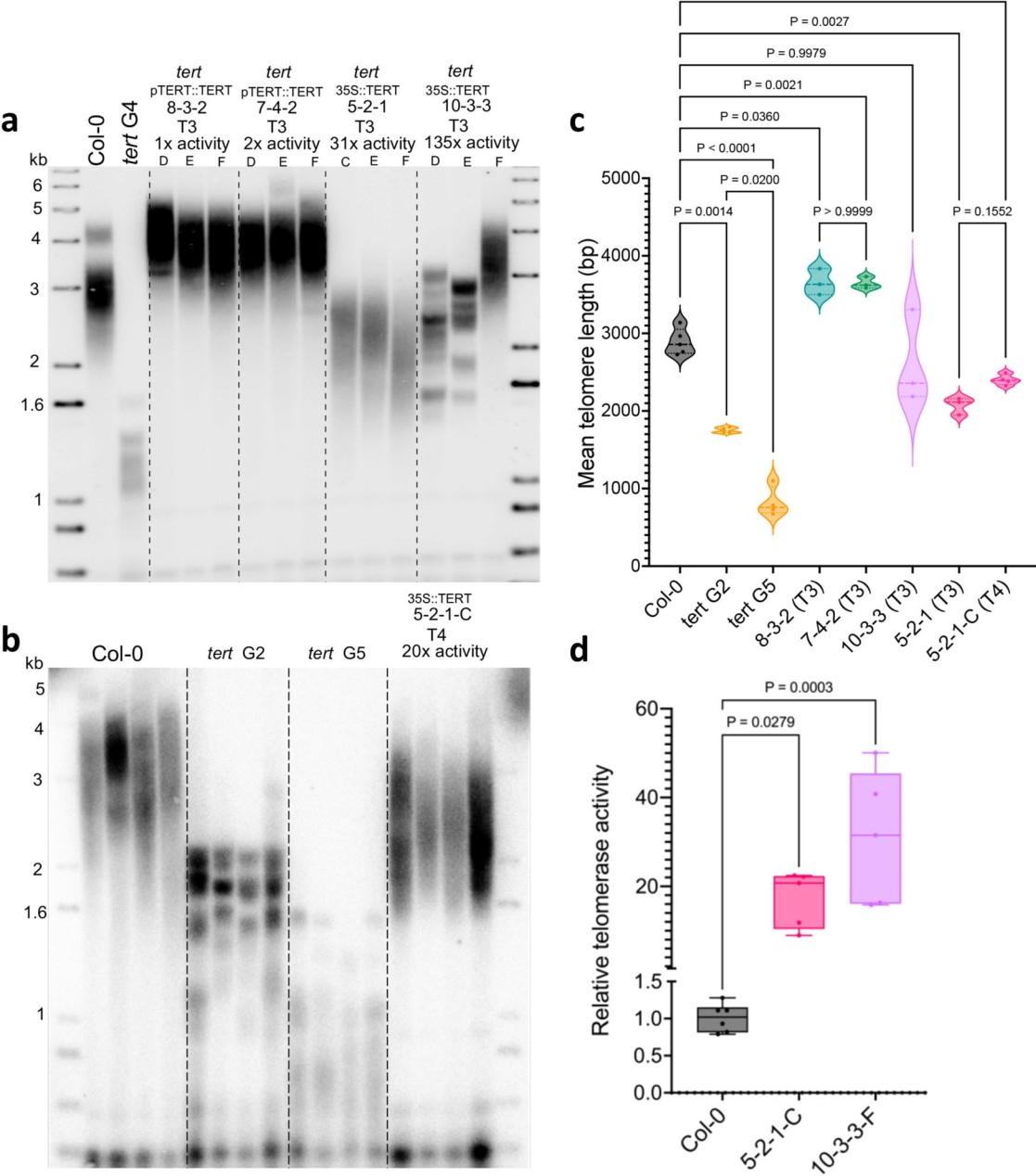

**Fig. 4 | Increased telomerase activity does not lead to telomere length changes.** *A. thaliana* plants genetically engineered to over-express telomerase were analyzed for telomere length. **a** TRF results for Col-0 ($n = 1$), G4 *tert* ($n = 1$), T3 telomerase-complementation lines 8-3-2 ($n = 3$) and 7-4-2 ($n = 3$) and T3 super-telomerase lines 5-2-1 ($n = 3$) and 10-3-3 ($n = 3$). Each lane shows the data for an individual biological replicate of the genotype indicated. Q-TRAP performed with flower bundles from T2 plants indicated that lines 8-3-2 and 7-4-2 express 1x and 2x the wildtype levels of telomerase activity, respectively, while 5-2-1 and 10-3-3 express 31x and 135x the wildtype levels of telomerase activity, respectively (see Supplementary Fig. 4). **b** TRF results for biological replicates of Col-0 ($n = 4$), G2 *tert* ($n = 4$), G5 *tert* ($n = 4$), and the T4 5-2-1-C line ($n = 4$). Q-TRAP performed with flower bundles from 5-2-1-C

indicated an ~20-fold increase in telomerase activity compared to wildtype (**d**). **c** Mean telomere length analyzed by WALTER for Col-0, G2 *tert*, G5 *tert*, T3 8-3-2, T3 7-4-2, T3 10-3-3, T3 5-2-1, and T4 5-2-1-C (data analyzed from Fig. 4a, b). Results shown as violin plots with *p* calculated as Brown-Forsythe ANOVA. **d** Relative telomerase activity measured by Q-TRAP of T4 super-telomerase lines 5-2-1-C ($n = 5$) and 10-3-3-F ($n = 5$) relative to the wildtype Col-0 control ($n = 6$). Technical replicates performed in triplicate. Within the boxplot, the middle line represents the median, box boundaries signify 25th and 75th percentiles, and whiskers the lowest and highest values. *P* calculated as one-way ANOVA, $n_{\text{Col-0}} = 6$, $n_{\text{5-2-1-C}} = 5$, $n_{\text{10-3-3-F}} = 5$, $p_{\text{5-2-1-C}} = 0.0279$ and $p_{\text{10-3-3-F}} = 0.0003$. Source data provided as Source Data File.

interaction between telomerase activity and organellar DNA abundance in Arabidopsis under normal conditions or upon stress.

We next asked if over-expression of telomerase could modulate the impact of oxidative stress in *A. thaliana* by measuring the endogenous levels of genomic 8-oxoG in T4 super-telomerase plants and in G2 and G5 *tert* mutants. For this experiment we extracted total DNA from 5-week-old plants and determined accumulation of 8-oxoG

relative to Col-0 plants ($n_{\text{Col-0}} = 20$). 8-oxoG was higher in G5 *tert* mutants (~1.7-fold, $n = 6$) than Col-0, while super-telomerase lines had significantly lower 8-oxoG accumulation than Col-0, with line 5-2-1-C (~0.75-fold, $n = 15$, $p < 0.0001$ by Brown-Forsythe and Welch ANOVA tests) higher than line 10-3-3-F (~0.44-fold, $n = 13$, $p < 0.0001$ by Brown-Forsythe and Welch ANOVA tests; Fig. 5d). Strikingly, direct comparison of relative telomerase activity and levels of genome oxidation in

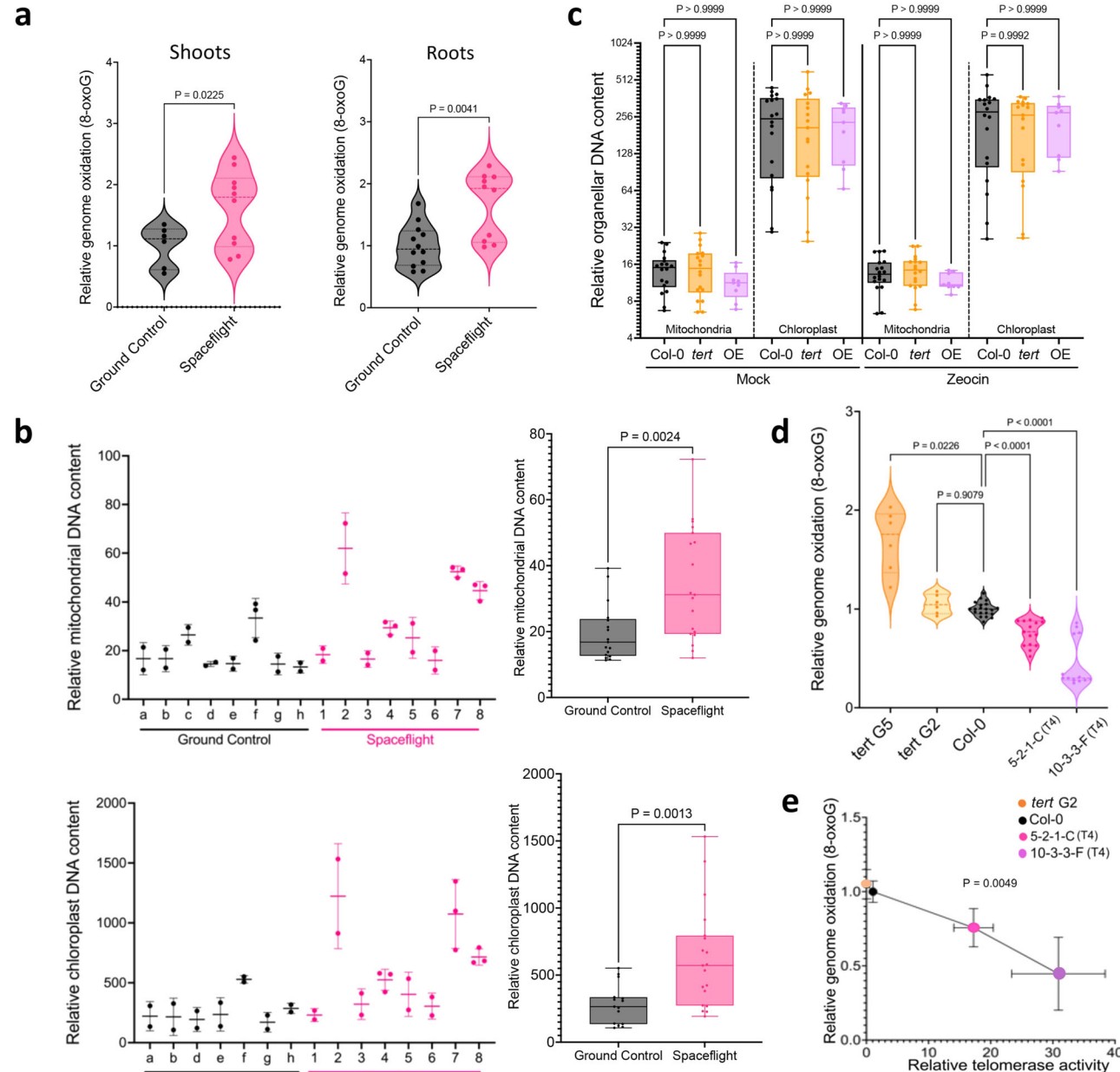

**Fig. 5 | Spaceflight leads to elevated nuclear 8-oxoG, and increased chloroplast and mitochondrial DNA. a** Genomic 8-oxoG content for shoots (left) and roots (right) (relative to total nuclear guanine content) in ground control ($n = 6$ shoots, $n = 12$ roots) and flight samples ($n = 10$ in shoots, $n = 10$ in roots). For each biological replicate, two technical replicates were performed. Results are shown in a violin plot; $p = 0.0225$ for shoots and $p = 0.041$ for roots calculated by unpaired two-tailed Welch's *t*-test. **b** Average change in abundance of mtDNA (top, left) and cpDNA (bottom, left) measured by qPCR in space-flown plants compared to ground controls for mitochondrial genes *cox1*, *atp1*, *nad*6 and *rps4* and chloroplast encoded genes *clpP*, *psbA*, and *ndhH* using *RpoTp* and *RpoTm* as nuclear controls. Combined relative changes in abundance of mtDNA (top, right) and cpDNA (bottom, right) in space-flown plants compared to ground controls are shown. The relative ratios of mtDNA to nuclear DNA were 20:1 for ground controls and 34:1 for spaceflight, and for cpDNA were 280:1 and 627:1, respectively. (left) Results are displayed as mean with *SD*, $n = 8$ for each organelle with three technical replicates per biological replicate for every organellar gene. (right) Within the boxplot, the middle line represents the median, box boundaries signify 25th and 75th percentiles and

whiskers, the lowest and highest values. Statistics calculated by unpaired two-tailed Welch's *t*-test, 17 total combined measurements from ground control and 19 total combined measurements from spaceflight, from 7 and 8 biological samples of each treatment, respectively, $p_{mitochondria} = 0.0024$, $p_{chloroplast} = 0.0013$. **c** Relative mitochondrial and chloroplast DNA measured by q-PCR in shoots of Col-0, G2 *tert*, and super-telomerase line 10-3-3-F. Six biological replicates per genotype with 18 total measurements per organelle. Within the boxplot, the middle line represents the median, box boundaries signify 25th and 75th percentiles and whiskers, the lowest and highest values. *P* calculated by one-way ANOVA. **d** Relative 8-oxoG content for 5-week-old plants for Col-0 ($n = 20$), G2 *tert* ($n = 6$), G5 *tert* ($n = 6$), T4 super-telomerase lines 5-2-1-C ($n = 15$) and 10-3-3-F ($n = 13$). For each biological replicate, two technical replicates were performed. *P* calculated by Brown-Forsythe and Welch ANOVA tests. **e** Direct comparison of relative telomerase activity and levels of genome oxidation in G2 *tert*, Col-0, and the two super-telomerase lines revealed a significant inverse correlation. Two-tailed Pearson's correlation, $r = -0.9951$, xy pairs ($n$) = 4, $p = 0.0049$. Source data provided as Source Data File.

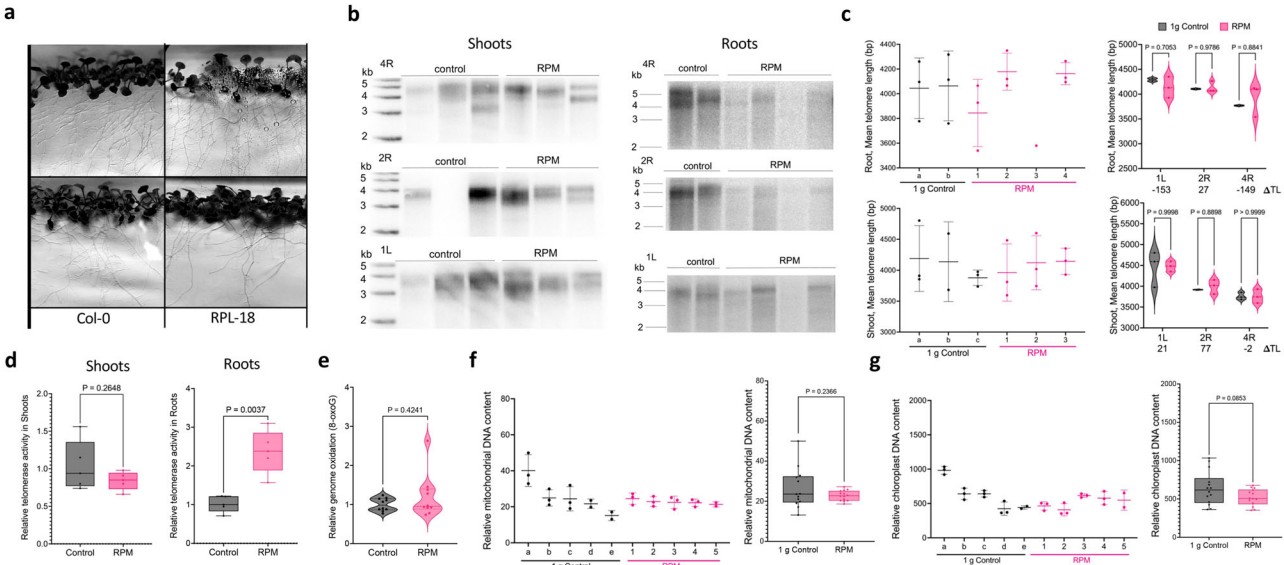

**Fig. 6 | Simulated microgravity does not recapitulate the telomerase and oxidative stress phenotypes of space-flown Arabidopsis. a** Photo of 12-day-old *35S::RPL18* and Col-0 seedlings grown on an RPM. **b** PETRA results for 12-day-old *35 S::RPL18 A. thaliana* seedlings grown in an RPM or in $1g$. Chromosome arms 5 R, 2 R and 1 L from pooled shoots (left) and roots (right) were analyzed; $n = 3$ for both ground control and RPM shoots. For roots, $n = 2$ for ground controls, and $n = 4$ for RPM samples. **c** (left) Telomere length for each biological root (left, top) and shoot (left, bottom) samples shown as mean and *SD* from chromosome arms in panel **b** and analyzed with WALTER. (right) Combined mean telomere length for each chromosome arm from root (right, top) and shoot samples (right, bottom) analyzed by WALTER with relative telomere length changes in base pairs (ΔTL) indicated. Results displayed as violin plot showing all biological points, and *p* calculated by two-way RM ANOVA. **d** Q-TRAP results for control and RPM-grown *35S::RPL18* roots (left) and shoots (right). For $1g$ controls, $n = 5$ and for RPM $n = 5$. For each biological replicate, three technical replicates were performed. Within the boxplot, the middle line represents the median, box boundaries signify 25th and 75th percentiles, and whiskers the lowest and highest values. $P = 0.2648$ for shoots and $p = 0.037$ for roots calculated by unpaired two-tailed Welch's *t*-test. **e** Genomic 8-oxoG (relative to total genomic guanine content) measured by ELISA. Results for RPM shoots and $1g$ *35 S::RPL18* control shoots are shown. For biological replicates, $n = 10$ for both $1g$ controls and RPM samples. Two technical replicates were performed for each biological replicate. Two-tailed Mann-Whitney test, $p = 0.4241$. **f** (left) Relative abundance of mtDNA measured by qPCR in RPM and $1g$ control plants. Each individual is shown as mean and *SD* from multiple mitochondrial markers (*cox1*, *atp1*, *rps4*) relative to nuclear DNA markers (*RpoTp* and *RpoTm*). (right) Relative abundance of mtDNA in RPM. $P = 0.2366$ by unpaired two-tailed Welch's test, 13 total combined measurements from $1g$ control and 14 total combined measurements from RPM, obtained from $n = 5$ in both treatments. Within the boxplot, the middle line represents the median, box boundaries signify 25th and 75th percentiles, and whiskers the lowest and highest values. **g** (left) Relative abundance of cpDNA measured by qPCR in RPM and $1g$ control plants. Each individual is shown as mean and *SD* from multiple chloroplast markers (*clpP*, *psbA*, and *ndhH*) relative to nuclear DNA markers (*RpoTp* and *RpoTm*). (right) Relative abundance of cpDNA in RPM. $P = 0.0853$ by unpaired two-tailed Welch's test, 14 total combined measurements obtained from $n = 5$ in both treatments. Within the boxplot, the middle line represents the median, box boundaries signify 25th and 75th percentiles, and whiskers the lowest and highest values. Source data provided as Source Data File.

G2 *tert*, Col-0, and the two super-telomerase lines revealed a significant inverse correlation (two-tailed Pearson correlation, xy pairs = 4, $p = 0.0049$) (Fig. 5e). This finding is consistent with a role for the telomerase enzyme in protection from genome oxidation.

High levels of oxidative damage in G5 *tert* mutants may be attributed in part to telomere dysfunction, since telomeres in our G5 *tert* mutants, but not in G2 *tert*, fell below the critical length threshold of 1 kb in length[8] (Fig. 4b, c). Moreover, prior transcriptomic studies of *tert* mutants show induction of the stress response in late generations, but not in G2 *tert*[78], correlating with the dramatic loss of telomere tracts in the former. Our model predicts that G2 *tert* mutants would have higher 8-oxoG content than wildtype, but we did not observe a significant change in 8-oxoG in G2 *tert* mutants ($n = 6$, $p = 0.9079$ determined by Brown-Forsythe and Welch ANOVA; Fig. 5d). It is possible that genome oxidation is actually greater in meristematic tissues of G2 *tert* mutants where telomerase is naturally most highly expressed, but since we analyzed DNA from whole plants this niche was not enriched. Future investigation of the spatial distribution of ROS and 8-oxoG in relation to local telomerase activity may prove enlightening in this regard.

**Simulated microgravity is insufficient to trigger the same molecular phenotypes as spaceflight**

Microgravity is one of the most conspicuous differences between Earth and low Earth-orbit. To explore mechanisms underlying the phenotypes observed in space-flown plants, seedlings of *35S::HF-RPL18*

(APEx-07 line) were grown for 12 days in simulated microgravity in a Random Positioning Machine (RPM)[79], which continually rotates biological materials in two different axes thereby randomizing Earth's gravity vector over time. As expected, the roots of RPM-grown seedlings did not clearly follow a gravitation vector (Fig. 6a). Similar to space-flown plants, PETRA analysis showed no difference in telomere length between plants grown at $1g$ or in simulated microgravity in the RPM, regardless of tissue (shoot or root) or chromosome arm (two-way RM ANOVA; Fig. 6b, c and Supplementary Fig. 7a, b). Average telomere length ($n = 3$, three chromosome arms) from the shoots of $1g$ control samples was 4060 bp, while RPM shoot telomeres averaged 4076 bp ($n = 3$, three chromosome arms), for a net difference of 16 bp ($p = 0.9322$ by unpaired two-tailed Welch's *t*-test, $n_{1g} = 8$, $n_{RPM} = 9$; Fig. 6b, c and Supplementary Fig. 7a, b). Root average telomere length ($n = 2$, three chromosome arms) was 4054 bp in $1g$ control and 4014 bp in RPM ($n = 4$, three chromosome arms), for a net difference of −40 bp ($p = 0.7578$ by unpaired two-tailed Welch's *t*-test, $n_{1g} = 6$, $n_{RPM} = 10$; Fig. 6b, c and Supplementary Fig. 7a, b).

Telomerase activity levels were unchanged in RPM shoots compared to $1g$ controls (unpaired two-tailed Welch's *t*-test, $n_{control} = 5$, $n_{RPM} = 5$, $p = 0.2648$; Fig. 6d, left). In contrast, there was a small but significant increase in telomerase activity in RPM roots of ~2.5 fold relative to $1g$ controls (unpaired two-tailed Welch's *t*-test, $n_{control} = 5$, $n_{RPM} = 5$, $p = 0.0037$; Fig. 6d right), however this telomerase induction was much lower than in space-flown plants. In addition, we found no

significant change in 8-oxoG measurements for RPM-grown shoots (two-tailed Mann-Whitney test, $n_{control} = 10$, $n_{RPM} = 10$, $p = 0.4241$; Fig. 6e) or the ratios of mtDNA and cpDNA relative to nuclear DNA compared to controls (unpaired, two-tailed Welch's t-test, 13 total combined measurements from 1 g control and 14 total combined measurements from RPM, from 5 biological samples of each treatment, respectively, $p_{mitochondria} = 0.236$, $p_{chloroplast} = 0.082$; Fig. 6f, g and Supplementary Fig. 7c).

Whereas the RPM is considered a better proxy than other microgravity simulators because the biological sample is subject to random rotations with no particular directional bias, it may not fully simulate microgravity conditions encountered in space[79]. Nevertheless, our findings suggest that other or combined stressors associated with spaceflight have a more profound impact on telomerase activity regulation and oxidative damage than microgravity alone. We postulate that space radiation is a strong contributing factor. Radiation exposure is one of many environmental assaults that increases intracellular ROS in plants[80]. Finally, we note that super-telomerase experiments conducted with both cancer and primary human cells demonstrated continuous telomere elongation in cells that over-express telomerase, with telomere tracts extending up to 8-fold longer than in controls[81]. Since telomerase activity is limiting for telomere extension in human cells, it is conceivable that telomere elongation in astronauts is a by-product of a telomerase-mediated mechanism to protect against ROS.

In summary, our findings provide unanticipated insight into telomere dynamics and regulation in Arabidopsis in space and on Earth. In addition to evidence of an oxidative stress protective function for telomerase that is uncoupled from telomere length maintenance, we highlight the remarkable adaptation of this organism to preserve its telomeres in a multitude of adverse settings. These achievements, along with the proficiency of plants more broadly to manage the oxidative stress created by high-energy photosynthesis reactions and the ever-changing environmental conditions they endure here on Earth, may help to sustain them through the extreme stresses imposed by long-duration space travel and, ultimately, interplanetary colonization.

## Methods

### Plant growth conditions and harvest
*A. thaliana* seeds (*35S::HF-RPL18* in the Columbia-0 background; ABRC stock # CS66056)[36] were adhered to a polyethersulfone (PES) membrane using guar gum[81]. Fifteen seeds were planted on each membrane. Seeded PES membranes were transferred to square 10 cm 0.5x Murashige and Skoog plates with 1% agar. Plates were kept in the dark at 4 °C for 19 days prior to integration into the Veggie growth chambers on the ISS and at Kennedy Space Center. Plants were grown for 12 days under 16 h/8 h long day conditions using 100 mol/m²/s of red (630 nm), blue (455 nm) and green (530 nm) light as is available in Veggie units. Ground control plates were seeded, grown and harvested at a 48-h delay from the flight samples. Continuous data on temperature, $CO_2$ and relative humidity were sent from ISS and emulated in the ground control environmental chamber on a 48-h delay. The average temperature was 23 °C with 44% relative humidity, and $CO_2$ levels were typically near 1750 ppm. After plates were removed from the Veggie, astronauts or ground control staff used forceps to collect the PES membranes and seedlings. Membranes were placed into the lid of the Petri dish for protection and wrapped with foil, then transferred to a −130 °C cold bag for rapid freeze. Seedlings were stored at −80 °C, shipped on dry ice and stored at −80 °C until processing.

Simulated microgravity treatments were carried out at NASA's Microgravity Simulation Support Facility (Kennedy Space Center, FL). RPMs (Airbus Defense and Space, Netherlands) were used to simulate microgravity and 1 g controls were conducted under static conditions. Environmental parameters (light, temperature and $CO_2$) were matched to Veggie conditions on the ISS. Plate setup, growth, harvest and shipping followed procedures of the ISS-grown seedlings. Plates were stratified for 5 days in the dark at 4 °C then placed onto RPMs or a 1 g control setup. Seedlings were grown under a 16 h/8 h light cycle for 12 days. Seed membranes were removed from plates, transferred to the plate lid, wrapped in foil, placed in a −130 °C cold bag, stored at −80 °C, then shipped on dry ice.

### Telomerase complementation and super-telomerase plants
Super-telomerase (*pHSN6A01 U6::AtTR 35S::TwinStrep tag II-TERT*) and complementation-telomerase (*pHSN6A01 U6::AtTR P_{TERT}::TwinStrep tag II-TERT*) plasmids were created by cloning *AtTR* downstream of the *U6* promoter in pHSN6A01 using NEBuilder® HiFi DNA with three fragments: the backbone was amplified in two fragments with primers (F1-f 5′- taagctcacgtgacggaattaagctCGACTTGCCTTCCGCACAATAC-3′ and F1-r 5′-tcccacaccccttATCACTACTTCGACTCTAGCTGTATATAAAC-3′, and F2-f 5′-ctcccaccccaaatattttTTTTGCAAAATTTTCCAGATCG-3′ and F2-r 5′-acctgcaggcatgcaagcttATTGGTTTATCTCATCGGAACTG CAAAAG-3′) and *AtTR* was amplified from Col-0 genomic DNA with primers (TR-f 5′-tcgaagtagtgatAAGGGGTGTGGGAACCTAG-3′ and TR-r 5′-cgatctggaaaattttgcaaaAAAATATTTGGGGGTGGGAG-3′). The *pHSN6A01 U6::AtTR* backbone was linearized with *Xba*I and *Sac*I and assembled with TwinStrep tag II and TERT fragments using NEBuilder® HiFi DNA to make the overexpression plasmid, *pHSN6A01 U6::AtTR 35S::TwinStrep tag II-TERT*. The TwinStrep tag II was amplified from the plasmid pTD-NTwin-St with primers (TS-f 5′-caatacttgtatggccgcgg ccgctCTAGATGGCTAGCGCTTGGAG-3′ and TS-r 5′-gtttacgcggcatCG AATTCGGGACCGCGGT-3′), and the telomerase gene was amplified from Col-0 genomic DNA with primers (TERT-f 5′- ggtcccgaattcgA TGCCGCGTAAACCTAGAC-3′ and TERT-r 5′-tccccaatacttgtatggaggcctg AGCTCTCAATAATTCAACTTCCAC-3′).

To make the complementation-telomerase lines, we constructed a complementation vector, *pHSN6A01 U6::AtTR P_{TERT}::TwinStrep II-TERT*, wherein the CaMV 35S promoter in the overexpression plasmid was replaced with the 2.1 kb sequence upstream of the *AtTERT* gene in Col-0. The overexpression plasmid was digested with *Sph*I and *Nhe*I, and the 1.1 kb band was purified for further assembly. $P_{TERT}$ was amplified from Col-0 genomic DNA with primers (PTERT-f 5′-TGAGATA AACCAATAAGCTTGCATGCAACCATGCATACCATAACCC-3′ and PTER T-r 5′-TGCGGGTGGCTCCAAGCGCTAGCCATTACACCTTCCTCCTCCT TTCT-3′).

To obtain telomerase complementation and super-telomerase plants, third generation telomerase reverse transcriptase null mutants, G3 *tert*, were transformed by floral dipping with *Agrobacterium tumefaciens* GV3101 carrying a binary vector for overexpression (*pHSN6A01 U6::AtTR 35S::TwinStrep tag II-TERT*) or complementation (*pHSN6A01 U6::AtTR P_{TERT}::TwinStrep tag II-TERT*), and the helper plasmid, pSOUP. T1 transgenic seedlings were selected in half strength MS media supplemented with 1% sucrose and hygromycin. Single copy transgenic lines were selected in T2 and assessed for complementation of telomerase activity Q-TRAP and telomere length by TRF. T3 and T4 homozygous single copy transgenic lines were used for follow-up experiments.

### Statistical analysis
All the statistical analyses were performed using the statistical package GraphPad Prism 9 for MacOS. All experiments were simultaneously performed on at least three biological samples (with the exception of TRAP analysis for space-flown roots as material was limited). In the case of all PCR and ELISA-based assays there was a minimum of three technical replicates. Differences were considered significant when two-tailed p-values were below 0.05. The specific tests used for each assay are cited alongside the data.

### Analysis of telomere length
Bulk telomere length was assessed using TRF analysis conducted with the *Mse*I restriction enzyme (New England Biolabs, Ipswich, MA, USA).

Single telomere analysis was performed using PETRA[8]. Initial primer extension was conducted with a primer that binds to the G-overhang region of telomeres (PETRA-T 5′-CTCTAGACTGTGAG ACTTGGACTACCCTAAACCCT-3′). PCR amplification employed a subtelomeric primer specific to the chromosome of interest (1R 5′-CTA TTGCCAGAACCTTGATATTCAT-3′; 1L 5′-AGGACCATCCCATATCATTG AGAGA-3′; 2R 5′-CAACATGGCCCATTTAAGATTGAACGGG-3′; 3R 5′-CT GTTCTTGGAGCAAGTGACTGTGA-3′; 3L 5′-CATAATTCTCACAGCAGC ACCGTAGA-3′; 4R 5′-TGGGTGATTGTCATGCTACATGGTA-3′; 5R 5′-CA GGACGTGTGAAACAGAAACTACA-3′; 5L 5′-AGGTAGAGTGAACCTAAC ACTTGGA-3′). A second primer (PETRA-A 5′-CTCTAGACTGTGAGAC TTGGACTAC-3′) was used in conjunction with the chromosome-specific primer during the PCR amplification step. TRF and PETRA blots were hybridized with a 5′ end [$^{32}$P] labeled $(T_3AG_3)_4$ probe. Telomere length on Southern blots was determined using the online tool WALTER[37]. Telomere-specific signals were converted to intensity profiles using the ScanToIntensity tool, and intensity profiles were assessed using the IntensityAnalyser tool. DNA size marker fitting with a 1 kb$^+$ ladder and selection of telomere-specific signal with background correction were performed to render statistical analysis. The qPCR-based telomere content assay was performed using the T/S method[10,39].

## Telomerase activity

To gauge telomerase activity, TRAP and Q-TRAP were performed[40]. For TRAP total protein was added to reactions containing 1x primer extension buffer (50 mM Tris-OAc pH 8.0, 50 mM KCl, 3 mM MgCl$_2$, 2 mM DTT, 1 mM spermidine), 0.66 μM forward primer (5′-CACTATC-GACTACGCGATCAG-3′), 0.83 mM dATP, 0.83 mM dTTP and 0.83 mM dGTP, which were kept at 37 °C for 45 min. The reaction mixture was subjected to phenol/chloroform extraction followed by overnight ethanol precipitation. After centrifugation at 15,000 rpm, 4 °C for 30 min, the pellet was washed with 70% ethanol and resuspended in nuclease-free H$_2$O. PCR reactions contained this template, GoTaq® Hot Start Colorless Master Mix (Promega), 0.4 μM forward primer (5′-CACTATCGACTACGCGATCAG-3′), 0.4 μM reverse primer (5′-CC CTAAACCCTAAACCCTAAA-3′) and 66 nM [α-$^{32}$P] dGTP-3000 Ci/mmol (PerkinElmer). After 35 cycles of PCR, the reaction was precipitated and resolved by 6% denaturing PAGE, dried, exposed and imaged on Typhoon FLA 9500 phosphorimager (GE Healthcare). For Q-TRAP, a protein extract dilution of 4.8 ng/μl (10.5 μl) was combined with 1 μl of 10 μM forward primer (5′-CACTATCGACTACGCGATCAG-3′) and 12.5 μl of Sybr Green PCR master mix (NEB). This mixture was incubated at 30 °C for 45 min on a qPCR plate. In all, 1 μl of the reverse primer (5′-CCCTAAACCCTAAACCCTAAA-3′) was introduced to the qPCR plate followed by 35 PCR cycles, involving a denaturation step of 30 sec at 95 °C, then annealing and extension steps of 90 sec each at 60 °C. Threshold cycle (Ct) values were calculated using an iCycler iQ thermal cycler (Bio-Rad). To ensure data accuracy, normalization to wildtype samples was performed.

## 8-oxo-G quantification

Genomic 8-oxoG was measured using a DNA Damage Competitive ELISA kit (Invitrogen™, ThermoFisher Scientific Cat. # EIADNAD)[10].

## Stress treatments

For light stress treatment[10] seeds were surface-sterilized in 2.7% sodium hypochlorite with 0.1% Triton X-100 for 10 min, stratified in 4 °C for 3 days, and plated on 0.8% agar plates with half-strength Murashige & Skoog (MS) minimum medium. Seeds were germinated under a 12-h photoperiod of an average illuminance of 5000 (±250) lux attained with 3000:6500 Kelvin lights in a 1:1 ratio, and at a constant temperature of 22 °C. In all, 7-day-old seedlings were transferred to soil (Sunshine, Mix 5, Sun Gro Horticulture, Agawam, MA, USA) and allowed to grow under the same conditions for 3 weeks. At this point,

half the plants were moved to a 24-h photoperiod chamber with an average illuminance of 8000 (±400) lux, maintaining the warm-to-cool light ratio. Two weeks later, whole plants from both light regimens were harvested for subsequent analyses.

Hydrogen peroxide measurements were conducted following a colorimetric method using potassium iodide[10,82]. Leaves from 4-week-old plants were homogenized with 1.5 ml of 0.1% (w/v) TCA, then centrifuged at 10,000×$g$ for 20 min at 4 °C. The supernatant (0.5 ml) was mixed with 1 ml of 1 M potassium iodide for 1 h in the presence of 0.5 ml of 0.1 M Tris-HCl (pH 7.6). The absorbance was read at 390 nm and hydrogen peroxide content was determined using a standard curve.

For zeocin treatment, seedlings were grown as described above. In all, 12-day-old seedlings were transferred to a medium containing 0.5 MS media with either mock or 20 μM zeocin (Invitrogen) treatment for 2 h. Roots and shoots were dissected for further analysis and immediately frozen on liquid N$_2$.

Additional methods can be found in Supplementary Methods of the Supplementary Information File.

## Reporting summary

Further information on research design is available in the Nature Portfolio Reporting Summary linked to this article.

## Data availability

Datasets analyzed for telomeric gene analysis from previous flights were retrieved from NASA's Genelab general data repository, which included GLDS-38[45], GLDS-120[47], GLDS-218[46], and GLDS-427[48]. The datasets generated during and/or analyzed during the current study are available from the corresponding authors upon request. Source data provided as a Source Data file. Source data are provided with this paper.

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

## Acknowledgements

This work was supported by the National Institutes of Health (R01 GM065383 to D.E.S), the National Aeronautics and Space Administration (80NSSC19K1481 to S.E.W. and I.P; 80NSSC23K0302 to D.E.S, S.E.W., and S.M.B; and NNX14AH51G, NNX14AB01G, and 80NSSC19K0434 to S.M.B.) and the NASA Postdoctoral Program at Kennedy Space Center administered by Oak Ridge Associated Universities (to A.M.). Special thanks to the APEx-07 team, especially Gerard Newsham, Anne Marie Campbell, Erica Bugardner, and Susan Manning-Roach, at Kennedy Space Center for their work with experiment validation and flight pre-paration and return, and astronauts Thomas Pesquet, Mark Vande Hei and Megan McArthur for experiment take down and harvest on ISS. Additional thanks to Maria Blasco for stimulating discussion and to the NASA Genelab Plant Analysis Working Group, the team at the NASA Microgravity Simulation Support Facility and Jeffrey Richards for accommodating the microgravity treatments.

## Author contributions

All authors contributed significantly to this work. B.B.B., A.D.M., C.C.G., P.Y., J.H.M., J.S., E.L., I.Y.P., S.M.B., R.A., S.E.W., and D.E.S. designed the experiments. B.B.B., A.D.M., C.C.G., P.Y., J.H.M., C.P., J.S., E.C., and E.L. performed experiments and analyzed data. B.B.B., A.D.M., C.G.G., D.E.S., and S.E.W. wrote the paper with contributions from all other authors.

## Competing interests

The authors declare no competing interests.
