## [Peer Review File · Nature Communications]

Arabidopsis telomerase takes off by uncoupling enzyme activity from telomere length maintenance in spaceEditorial Note: This manuscript has been previously reviewed at another journal that is not operating a transparent peer review scheme. This document only contains reviewer comments and rebuttal letters for versions considered at *Nature Communications*. Mentions of the other journal have been redacted.

REVIEWER COMMENTS

Reviewer #1 (Remarks to the Author):

First, I would like to thank the Authors for carefully addressing most of my comments to the original version sent to [redacted]. In a result, the technical level of the MS has considerably improved in the revised version. I understand that in some cases (e.g., point 7), this was not possible due to the limited availability of the material, and mainly due to the inappropriate design of the experiment for the purpose of this study. For example, the use of tert mutants as one of samples in the spaceflight plant material could clarify the otherwise problematic interpretation of the results. Importantly, the observed discrepancy is not explained between the enormous increase in telomerase activity on one hand, and no increase in expression of telomerase core subunits (TERT and TR) on the other hand (see below), due to the absence of appropriate experiments. The follow up ground experiments have been designed and performed carefully but their relevance to spaceflight experiment is questionable. The observed possible role of telomerase and its components in protection against oxidative DNA damage concurs with previous reports on non-telomeric roles of telomerase, and is not surprising. Unfortunately, even this part suffers from limitations of the spaceflight experiment (e.g., oxidative damage of nuclear, mtDNA and cpDNA have not been analysed separately, and therefore, any direct comparison with previous studies cannot be performed to formulate a more precise and experimentally supported mechanistic explanation).

Major criticisms:

1. L 169-172 (and similar statements throughout the MS): The sentences starting with "Contrary to astronaut expression studies..." – First, the comparison between telomere/telomerase measurements in humans and plants is misleading: while in humans, peripheral leukocyte telomere length is usually measured, i.e. differentiated somatic cells reflecting not only TL in bone marrow progenitor cells but also loss of TL during further proliferation and differentiation, already without telomerase, in plant seedlings the authors deal with a mixture of dividing meristem cells and non-dividing differentiated cells in unknown ratio. I understand the authors aim to attract attention to their study (therefore, they extensively summarize the studies on astronauts in the Introduction, lines 73-90). However, due to key differences between plant and human individual development, stem cells and somatic cells would have to be analysed separately.

2: Even more importantly, the discrepancy between the unchanged levels of expression of telomerase subunits, and highly increased telomerase activity is "explained" by posttranscriptional control. However, this explanation is too vague on one hand, and only one of many possible explanations. E.g., a simple microscopic analysis of root anatomy in seedlings could show whether the meristematic zone is expanded (e.g., due to microgravity conditions) – and therefore, the change reflects the increased meristem cell number, rather than an increase telomerase activity per cell. And this is the key question of the study, in my view.

Minor points:

1. In the Abstract, please, add "in roots" (original Minor point 1, reword to "up to 150-fold in roots").

2. Abstract, lines 39-40: The last sentence of the Abstract could be omitted or reworded, as it says nothing important in addition what is said by the previous sentence (We propose...) which could be the last sentence of the Abstract – highlighting the putative role of telomerase as a protection against DNA oxidation. Telomerase is essential for a long-term survival of any organism, both on Earth and in

space – in this respect, the study did not bring any novel support/insight.

3. Introduction, L55-57. The Authors cite paper on Ku mutants (Zellinger et al., 2007). However, the problems of Ku mutants are rather caused by the deficiency in Ku, and its role in protection against alternative telomere lengthening, formation of telomeric circles etc..., than by the increased telomere length itself.

4. L 151: The paragraph title - remove the second "increase" - Spaceflight leads to a dramatic increase in telomerase activity increase in Arabidopsis roots

Reviewer #2 (Remarks to the Author):

Nat Comm "Telomerase takes off"

Summary:

The authors here attempt to determine whether spaceflight alters the length of telomeres and the activity of telomerase, as has been observed previously in a limited sample of astronauts (3 out of 4 displayed telomere lengthening in peripheral blood mononuclear cells, and a similar effect was suggested, though not statistically supported, in a *C elegans* study). Imbibed seeds on agar plates were sent up to the space station, where they germinated and grew over a period of 12 days, with control seeds in a similar chamber on Earth. The authors then separated root from shoot to analyze telomere length, telomerase activity, and 8oxoG concentration in roots vs shoots. They find no significant change in telomere length, a large and significant upregulation of telomerase activity, and a slight but significant increase in 8oxoG in both roots and shoots in space vs ground-based seedlings. To further investigate the apparent lack of correlation between telomerase expression and telomere length (as well as correlations with oxidative damage), the authors build telomerase overexpression lines, driving telomerase with a viral 35S promoter. This choice of promoter is unfortunate, as detailed below, as it is highly expressed in mature tissues, but thought to be expressed weakly in embryonic and meristematic tissues- how this weak expression compares with normal expression from the native promoter is unclear. They transform this construct into a telomerase KO line and assay telomere length and telomerase activity after 4 and 5 sexual generations, finding that telomeres are not longer in the OE lines. This provides a second example of (like the spaceflight experiment) of an apparent discordance between telomere length and telomere expression, but see below. This suggests that Arabidopsis telomere length is tightly regulated in a manner that doesn't depend on telomerase expression. This would be an interesting observation but there are some issues with the experimental design that lead me to question this conclusion.

The authors also suggest that telomeres, or telomerase expression, has a role to play in ROS homeostasis, a topic that is also discussed in the mammalian literature, as they see a correlation between 8oxoG levels and telomerase expression levels (but not telomere length).

My major issues:

Given the relatively short duration (12 days) of this experiment (vs the months-long astronaut observation), and the population of cells assayed (the majority of which do not normally express telomerase, and are established prior to germination) this comparison between Arabidopsis seedlings and human RBMCs (not "astronauts"- just astronaut RBMCs) is not valid. The human studies assayed an effect, over several months, on a population of relatively short-lived cells, cells that have a documented ability to "switch on" telomerase. The seedling assay, in contrast, was over 12 days (not months), in whole plants, which include many cell types, the majority of which do not express telomerase. This affects the ability of the assays to detect correlations between telomere length and telomerase expression. For example, if, as in ground-based plants, all the telomerase (enzyme) signal in a root is coming from a small subset of cells (the meristem, perhaps 1/100 cells in the roots of a 12 day old seedling?) and the expression of telomerase doubled in *just* those cells, that would still

register as a doubling of expression for "roots"- which is fine! But if the length of the telomeres doubled in this this same subpopulation, that change would be imperceptible, because the meristem's increased telomere length signal is swamped by the (perhaps unchanged) length signal of the majority of cells. That could explain the lack of effect on telomere length- it's an artifact of looking at whole roots, rather than just the cells producing telomerase.

Thus, although it is entirely possible that arabidopsis seedlings don't change their telomere length in space, no convincing evidence is provided to "prove" this negative. "Negative results"- defined as the lack of effect of a stimulus- ARE important, but this data doesn't prove there is no effect, due to the confounding effects of this particular mixed population of cells (and the perhaps the very short time provided). I would still present and discuss this data, but I would caution the reader against over-interpretation- and tell them why. I certainly wouldn't present "lack of lengthening" as a conclusion, or mention it in the abstract.

Similarly, the authors assay expression of telomerase in their "super telomerase expressors" in "flower bundle". I'm not sure what this term means but I'm guessing it is the tip of the inflorescence, containing the stem, pedicles, and many unopened buds. It's a good choice- better than seedling roots- if you want an easily harvested tissue enriched in mitotic and germline precursor cells, as well as the inflorescence meristem. They have chosen the 35S promoter because it is a strong promoter, but, unfortunately, it is only strong in mature, differentiated tissues. 35S is thought to be a weak promoter in embryonic or meristematic cells, which are the cells that contribute to the germline. Thus the source of the mismatch between telomerase expression and telomere length *might* simply be due to the fact that overexpression is occurring only in cell lineages that do not contribute to the next generation of plants. This would result in telomere length being reset, with every sexual generation, to reflect whatever level of expression is present in the apical/inflorescence meristems. In other words, the nonmeristematic tissue in the sample might be expressing high levels of 35S-driven telomerase (detected in the assay here), but the overall telomere length reflects the level of expression (with 35S, low) in the meristems.

Minor issues:

L21 "changes in astronaut telomeres" might as well be unambiguous- say "reduction in telomere length"

Abstract- again, I don't think this experimental design allows us to draw the conclusion that telomere length is unaffected.

L57 add "at least in a ku defective background". Certainly telomere instability would both be enhanced and have more dire consequences in a ku background- long telomeres in a wt background might be harmless.

L94 again, it's not clear there was time to see an effect on telomere length- the time scale difference should be mentioned, and the human tissue assayed should be mentioned.

L105 how old are the seedlings in these images? Add this to legend? Also, it's really hard to see that these are agravitropic- the photos mostly give the impression of spaceflight plants producing very few primary or secondary roots. Otherwise the angle of growth doesn't seem very different. Planting these seeds in the middle of the plate instead of the top might have helped- we can't see the roots that grow "up" in these photos.

L168 "than what we can report here" not clear to me what this means

L176 ref 51 refers to an oxidative burst that occurs when the lights are turned out after an extended (36 hr) day, in plants previously grown under short day conditions. That's not what you're doing here- you're doing continuous light. I'd drop this ref. Can you find another ref- besides 10- that supports the notion that continuous light is a stressor? There are no obvious reasons why it should be.

L184 I don't know what flower bundles are

L207 + 221 where's the T2 data and statistical justification?

L225 not clear why this promoter was chosen or why the overexpressor wasn't built into wt instead of tert, please explain rationale. Also telomere length in the OE line *looks* significantly shorter than wt here. Instead of saying that this is "within the normal range" please do a statistical analysis.

L237 P values don't match those in the figure, or I'm just tired...

L286 telomerase, but not telomeres, right? Make your model could be made a little more clear?

L366 in methods, I'm guessing seeded membranes (= time of seed imbibition) were prepared on the ground. Please clarify how long the seeded plates were left in the cold (what temp?) and dark, and if ground-based controls experienced a longer stay in the cold and dark and give us the temperature and % CO₂ used.

In Conclusions, please provide some more specific conclusions- specific things that we learned from this work.

Reviewer #3 (Remarks to the Author):

The authors addressed all of my comments and suggestions. Furthermore, they added new data showing that oxidative damage induced by light stress or a radiomimetic drug led to the upregulation of telomerase. They also found that plants overexpressing telomerase had lower levels of oxidative damage, as assessed by the accumulation of 8-oxoG. These additional data provide further validation for the observations made using the unique material generated during space flight. While this study does not provide deeper mechanistic insights into how telomerase implements its protective function against oxidative damage, it still presents interesting observations on how space flight affects plant physiology.

I have a few remarks:

Line 162: In the reference to single-cell transcriptomics, the authors cite papers 43 and 44. However, these studies do not contain any single-cell transcriptomics data. The authors mention in parentheses that data were also analyzed from reference 45, but no further details are provided. This should be clarified.

Line 171: Post-transcriptional gene regulation usually refers to regulatory mechanisms between transcription and translation. However, the data in the paper do not exclude regulation at the level of translation, complex assembly, or post-translational modification.

Line 179: Units after numerical values should not be written in parentheses.

Figure 4 and corresponding text in the manuscript: The observation that overexpression of TERT from the 35S promoter does not fully rescue telomere length has already been made in Watson et al. (Genetics, 2021). These results should be discussed in the study.

RESPONSE TO REVIEWER COMMENTS

Reviewer #1 (Remarks to the Author):

First, I would like to thank the Authors for carefully addressing most of my comments to the original version sent to [redacted]. In a result, the technical level of the MS has considerably improved in the revised version. I understand that in some cases (e.g., point 7), this was not possible due to the limited availability of the material, and mainly due to the inappropriate design of the experiment for the purpose of this study. For example, the use of tert mutants as one of samples in the spaceflight plant material could clarify the otherwise problematic interpretation of the results. Importantly, the observed discrepancy is not explained between the enormous increase in telomerase activity on one hand, and no increase in expression of telomerase core subunits (TERT and TR) on the other hand (see below), due to the absence of appropriate experiments. The follow up ground experiments have been designed and performed carefully but their relevance to spaceflight experiment is questionable. The observed possible role of telomerase and its components in protection against oxidative DNA damage concurs with previous reports on non-telomeric roles of telomerase, and is not surprising. Unfortunately, even this part suffers from limitations of the spaceflight experiment (e.g., oxidative damage of nuclear, mtDNA and cpDNA have not been analysed separately, and therefore, any direct comparison with previous studies cannot be performed to formulate a more precise and experimentally supported mechanistic explanation).

Major criticisms:

1. L 169-172 (and similar statements throughout the MS): The sentences starting with "Contrary to astronaut expression studies..." – First, the comparison between telomere/telomerase measurements in humans and plants is misleading: while in humans, peripheral leukocyte telomere length is usually measured, i.e. differentiated somatic cells reflecting not only TL in bone marrow progenitor cells but also loss of TL during further proliferation and differentiation, already without telomerase, in plant seedlings the authors deal with a mixture of dividing meristem cells and non-dividing differentiated cells in unknown ratio. I understand the authors aim to attract attention to their study (therefore, they extensively summarize the studies on astronauts in the Introduction, lines 73-90). However, due to key differences between plant and human individual development, stem cells and somatic cells would have to be analysed separately.

We agree with the reviewer and have removed from the text all direct comparisons between plant and human data. However, the premise for the study is based on prior spaceflight research and therefore we retained in the introduction a summary what is known concerning the impact of spaceflight on telomeres in humans and in *C. elegans*.

2: Even more importantly, the discrepancy between the unchanged levels of expression of telomerase subunits, and highly increased telomerase activity is "explained" by posttranscriptional control. However, this explanation is too vague on one hand, and only one of many possible explanations. E.g., a simple microscopic analysis of root anatomy in seedlings could show whether the meristematic zone is expanded (e.g., due to microgravity conditions) – and therefore, the change reflects the increased meristem cell number, rather than an increase telomerase activity per cell. And this is the key question of the study, in my view.

We agree that there are many possible explanations for telomerase regulation in this setting. Since we are unable to address the mechanism, we modified the text to state, "The mechanism of telomerase regulation is unknown".

Minor points:

1. In the Abstract, please, add "in roots" (original Minor point 1, reword to "up to 150-fold in roots").

We apologize for the oversight, and we made the correction.

2. Abstract, lines 39-40: The last sentence of the Abstract could be omitted or reworded, as it says nothing important in addition what is said by the previous sentence (We propose...) which could be the last sentence of the Abstract – highlighting the putative role of telomerase as a protection against

DNA oxidation. Telomerase is essential for a long-term survival of any organism, both on Earth and in space – in this respect, the study did not bring any novel support/insight.

We combined the last two sentences of the abstract to state, “We propose a redox protective capacity for Arabidopsis telomerase that may promote survivability in harsh environments”.

3. Introduction, L55-57. The Authors cite paper on Ku mutants (Zellinger et al., 2007). However, the problems of Ku mutants are rather caused by the deficiency in Ku, and its role in protection against alternative telomere lengthening, formation of telomeric circles etc..., than by the increased telomere length itself.

The Campitelli study (ref 6), which we also cite, revealed that telomeres in *ku* mutants are susceptible to shortening in a dry environment, but not with heat stress. Because we cannot know whether this is related to Ku’s role in modulating telomere recombination or simply because telomeres are elongated, we softened the remainder of the sentence to state, “raising the possibility that ultra-long telomeres (at least in a *ku*-defective background) may be more fragile in response to stress”.

4. L 151: The paragraph title - remove the second “increase” - Spaceflight leads to a dramatic increase in telomerase activity increase in Arabidopsis roots

We made the correction.

Reviewer #2 (Remarks to the Author):

Nat Comm “Telomerase takes off”

Summary:

The authors here attempt to determine whether spaceflight alters the length of telomeres and the activity of telomerase, as has been observed previously in a limited sample of astronauts (3 out of 4 displayed telomere lengthening in peripheral blood mononuclear cells, and a similar effect was suggested, though not statistically supported, in a *C elegans* study).

Imbibed seeds on agar plates were sent up to the space station, where they germinated and grew over a period of 12 days, with control seeds in a similar chamber on Earth. The authors then separated root from shoot to analyze telomere length, telomerase activity, and 8oxoG concentration in roots vs shoots. They find no significant change in telomere length, a large and significant upregulation of telomerase activity, and a slight but significant increase in 8oxoG in both roots and shoots in space vs ground-based seedlings.

To further investigate the apparent lack of correlation between telomerase expression and telomere length (as well as correlations with oxidative damage), the authors build telomerase overexpression lines, driving telomerase with a viral 35S promoter. This choice of promoter is unfortunate, as detailed below, as it is highly expressed in mature tissues, but thought to be expressed weakly in embryonic and meristematic tissues- how this weak expression compares with normal expression from the native promoter is unclear. They transform this construct into a telomerase KO line and assay telomere length and telomerase activity after 4 and 5 sexual generations, finding that telomeres are not longer in the OE lines. This provides a second example of (like the spaceflight experiment) of an apparent discordance between telomere length and telomere expression, but see below. This suggests that Arabidopsis telomere length is tightly regulated in a manner that doesn’t depend on telomerase expression. This would be an interesting observation but there are some issues with the experimental design that lead me to question this conclusion.

The authors also suggest that telomeres, or telomerase expression, has a role to play in ROS homeostasis, a topic that is also discussed in the mammalian literature, as they see a correlation between 8oxoG levels and telomerase expression levels (but not telomere length).

My major issues:

Given the relatively short duration (12 days) of this experiment (vs the months-long astronaut observation), and the population of cells assayed (the majority of which do not normally express

telomerase, and are established prior to germination) this comparison between arabidopsis seedlings and human RBMCs (not "astronauts"- just astronaut RBMCs) is not valid. The human studies assayed an effect, over several months, on a population of relatively short-lived cells, cells that have a documented ability to "switch on" telomerase. The seedling assay, in contrast, was over 12 days (not months), in whole plants, which include many cell types, the majority of which do not express telomerase.

While it is true that the original astronaut studies spanned several months duration, the most recent 2021 SpaceX Inspiration4 mission (two manuscripts currently under review as part of the space paper collection for Nature) was only 3-days duration with telomeres sampled during and post-flight. For all crew members telomere extension was observed at the first time point taken, indicating that the telomere response is rapid in human cells. We also note that the telomere length changes observed in *C. elegans* were detected during an 11-day mission. Thus, the timeframe for telomere length analyses in the most recent astronaut mission and the *C. elegans* study were similar to our Arabidopsis study. Nevertheless, we concur that a direct comparison of plant telomeres and telomerase with humans (and *C. elegans*) is not valid. Therefore, we removed these comparisons from the text. See response to Major Criticism 1 from Reviewer 1.

This affects the ability of the assays to detect correlations between telomere length and telomerase expression. For example, if, as in ground-based plants, all the telomerase (enzyme) signal in a root is coming from a small subset of cells (the meristem, perhaps 1/100 cells in the roots of a 12 day old seedling?) and the expression of telomerase doubled in *just* those cells, that would still register as a doubling of expression for "roots"- which is fine! But if the length of the telomeres doubled in this this same subpopulation, that change would be imperceptible, because the meristem's increased telomere length signal is swamped by the (perhaps unchanged) length signal of the majority of cells. That could explain the lack of effect on telomere length- it's an artifact of looking at whole roots, rather than just the cells producing telomerase. Thus, although it is entirely possible that arabidopsis seedlings don't change their telomere length in space, no convincing evidence is provided to "prove" this negative. "Negative results"- defined as the lack of effect of a stimulus- ARE important, but this data doesn't prove there is no effect, due to the confounding effects of this particular mixed population of cells (and the perhaps the very short time provided). I would still present and discuss this data, but I would caution the reader against over-interpretation- and tell them why. I certainly wouldn't present "lack of lengthening" as a conclusion, or mention it in the abstract.

Similarly, the authors assay expression of telomerase in their "super telomerase expressors" in "flower bundle". I'm not sure what this term means but I'm guessing it is the tip of the inflorescence, containing the stem, pedicles, and many unopened buds. It's a good choice- better than seedling roots- if you want an easily harvested tissue enriched in mitotic and germline precursor cells, as well as the inflorescence meristem. They have chosen the 35S promoter because it is a strong promoter, but, unfortunately, it is only strong in mature, differentiated tissues. 35S is thought to be a weak promoter in embryonic or meristematic cells, which are the cells that contribute to the germline. Thus the source of the mismatch between telomerase expression and telomere length *might* simply be due to the fact that overexpression is occurring only in cell lineages that do not contribute to the next generation of plants. This would result in telomere length being reset, with every sexual generation, to reflect whatever level of expression is present in the apical/inflorescence meristems. In other words, the nonmeristematic tissue in the sample might be expressing high levels of 35S-driven telomerase (detected in the assay here), but the overall telomere length reflects the level of expression (with 35S, low) in the meristems.

We appreciate the reviewer's thoughtful comments concerning the cellular complexity of the root and the issues related to expression with a 35S promoter. We also agree that our experiments cannot formally rule out the possibility that spaceflight allows telomeres in a few cells of the plant to be extended beyond the wildtype size range. However, data from a variety of on-ground experiments with Arabidopsis support the hypothesis that telomerase activity is not limiting for telomere elongation, and hence it is reasonable to postulate based on multiple lines of evidence presented in this paper, previously published work and unpublished data we include below that spaceflight does not now obligately couple telomere extension to telomerase activity.

In the modified text, we explicitly state the caveats related to the possibility of a few cells extending their telomeres in space-flown plants, and the limitations of the 35S promoter experiments. We also present additional evidence supporting the hypothesis that telomere length in Arabidopsis is determined accessibility at the chromosome terminus rather than telomerase enzyme activity levels per se. These considerations are described below.

1) A recent publication from the Riha and Shippen labs showed that telomeres can be grossly extended in plants expressing a 35S-driven hypomorphic TERT allele (35S:TERT) if Ku, the blunt-end telomere capping protein, is mutated (Watson et al. 2021). With the hypomorphic allele, telomerase is very lowly expressed and these plants have very short telomeres. However, in the first plant generation after Ku is lost, there is unambiguous elongation of bulk telomeres that is obvious in a standard TRF blot (Figure 2 from Watson et al., 2021). Other studies confirm that telomere extension in *ku* mutants is telomerase-dependent. Therefore, because bulk telomeres can be markedly extended, even when telomerase levels are much lower than in wildtype, these findings indicate that 35S:TERT expression is not limiting for telomere extension.

2) To more directly address the issues related to expression from a 35S promoter, we generated plants expressing TERT from a ubiquitin promoter. We provide this information to the reviewer here, but because the data are preliminary and limited (with only one relevant transgenic line), we did not include it in the revised manuscript. Analysis of T3 (UBQ:TERT) transformants showed that while TERT mRNA is highly expressed in several transformants (Fig. 1a), telomerase activity is not strongly induced as when both TERT and TR are over-expressed (Fig. 1b and see Fig. 4 in the manuscript). Thus, TR along with TERT appears to be limiting for telomerase activity. Nevertheless, in one UBQ:TERT line (Q8-2-2-5) telomerase activity was 2.5 fold higher than wildtype (n=5, p<0.0001) (Fig. 1b). We analyzed telomere length in this line and the other transformants and found no change in telomere length compared to wildtype after three plant generations (Fig. 1c-e). These results are consistent with the conclusion that telomerase activity is not limiting for telomere extension in Arabidopsis.

Figure 1. Telomeres are not extended in plants expressing a ubiquitin-driven TERT allele. **a** Relative TERT expression measured in T3 12-day-old pooled seedlings by Q-PCR with Actin2 as a reference. Results are shown for pooled UBQ:TERT lines (*tert* background) Q8-2-2-1 (n=2), Q8-2-2-2 (n=1), Q8-2-2-4 (n=1), and Q8-2-2-5 (n=2). **b** Relative telomerase activity for pools of T3 seedlings of Col-0 (n=3), and UBQ:TERT (*tert* background) Q8-2-2-1 (total n=8), Q8-2-2-2 (n=5), Q8-2-2-4 (n=5) and Q8-2-2-5 (n=5). For each biological replicate, three technical replicates were performed. **c** PETRA results for 12-day old UBQ:TERT (WT Col-0 background) seedlings. Chromosome arms 5R, 2R and 1L from pooled seedlings were analyzed. Col-0 (n=3) and UBQ:TERT Q8-2-2-1 (n=2), Q8-2-2-2 (n=2), Q8-2-2-4 (n=3), Q8-2-2-5 (n=3). **d** PETRA quantification of the mean telomere length for each genotype in every chromosome arm displayed determined by WALTER. **e** Combined mean telomere length for each chromosome arm from all the seedlings analyzed by WALTER.

Methods: To make ubiquitin-driven TERT complementation constructs (pBA pUBQ::Myc-sfGFP11-TERT and PBA pUBQ::TERT-Myc-sfGFP11) we modified the binary vector pBA-Flag-4Myc-DC⁸⁰ using NEBuilder® HiFi DNA with five and six fragments, respectively. The plasmid backbone (2) consisted of the purified 9.3kbp band obtained from cutting pBA-

Flag-4Myc-DC with PpuMI and PacI; the ubiquitin promoter (pUBQ) and tag (Myc-sfGFP11) were amplified from the plasmid pCambia 1380 pUBQ10:mCherry-Myc-sfGFP11:NOST (Addgene stock 97395); and, the TERT gene and TERT 3'UTR fragments were amplified from Col-0 genomic DNA. The primers for the N-terminal tagged TERT construct are: (2) pUBQ-f 5'-cagcgtgaagccttgcctgagcgaattcaaaaattcggatgatgaatag-3' and pUBQ-M-r 5'-tttgctccatgaattcgctgcacatacataac-3'; (3) MG-f 5'-cagcgaattcATGGAGC AAAAGCTCATTCTGAAGAGGAC-3' and MG-r 5'-tctcggcgcTGTGATGCCGGC GGCGTT-3'; (4) gTERTi8-f 5'-cgcatcacaGGCGCCGAGACCGCGTC-3' and gTERTi8-r 5'-tccttgaactCTGTAATACAA GATATGCAAGGTTTCGCTCACTTCAAGGAAGCAC-3'; and (5) gTERTex9-f 5'-tgtattacagAGTTCAAGGAGGCAAAAAG-3' and gTERT-3UTR-r 5'-attcgagctcactagttaatTTAAATCTTTCTTACCCTAAATG-3'. The primers for the C-terminal tagged TERT construct are: (2) pUBQ-f and pUBQ-T-r 5'-tacgcggcatGAATTCGCTGCACATACATAAC-3'; (3) gTERT-f 5'-cagcgaattc ATGCCGCGTAAACCTAGAC-3' and gTERTin8-r; (4) gTERTex9-f and gTERTns-r 5'-gctttgctccatgcatgcTGCATAATTCAAC TTCCAC-3'; (5) link-MG-f 5'-aGCATGCATGGAGCAAAAGCTC and MG*-r 5'-atcagttcaTCATGTGATGCCGGCGGC-3'; and (6) TERT3UTR-f 5'-catcacatgaTGAACCTGATCTTAACTAGATTTTATC-3' and TERT-3UTR-r 5'-attcgagctcactagttaatTTAAATCTTT CTTTACCCTAAATG-3'.

3) Telomerase is not normally expressed in *A. thaliana* leaves. However, telomerase is "activated" in an activation tagged line mutant (over-expressing a zinc-finger transcription factor) which leads to constitutively expression of telomerase in leaves. In these plants telomeres in leaves are not extended beyond the wildtype length (Ren et al. 2004). In addition, a recent study of telomere dynamics in *A. thaliana* calli (Sovakova et al., 2018) found that telomere elongation (up to 2-fold) occurred with extended passages, but this was not correlated with increased telomerase activity. These studies provide further evidence that telomerase activity and telomere elongation are not obligately coupled in Arabidopsis.

4) In budding yeast there is a precedent that telomerase is not limiting for telomere extension (in contrast to the situation in human cells). Yeast telomeres are recalcitrant to telomerase-mediated extension unless chromosome ends are in an "extendable" chromatin conformation (Teixera et al., 2002, 2004). Indeed, it is now clear that a complex interplay between telomerase and telomere-associated proteins, DNA replication and repair machinery and the telomere epigenome establishes telomere length homeostasis in budding yeast (Lustig 2019). We postulate a similar mechanism operates in *A. thaliana*. The more accessible chromosome ends in ku mutants provide an explanation for why the hypomorphic 35S:TERT is able to drive bulk telomere elongation.

Minor issues:

L21 "changes in astronaut telomeres" might as well be unambiguous- say "reduction in telomere length"

Astronaut telomeres are elongated during spaceflight, but then they shorten below controls after return to Earth. Therefore, it is more accurate to describe this as "changes in telomere length".

Abstract- again, I don't think this experimental design allows us to draw the conclusion that telomere length is unaffected.

See response above.

L57 add "at least in a ku defective background". Certainly telomere instability would both be enhanced and have more dire consequences in a ku background- long telomeres in a wt background might be harmless.

We made the suggested change.

L94 again, it's not clear there was time to see an effect on telomere length- the time scale difference should be mentioned, and the human tissue assayed should be mentioned.

We added this information to the introduction. Also see the response above concerning time scales for the spaceflight experiments in astronauts and *C. elegans*.

L105 how old are the seedlings in these images? Add this to legend? Also, it's really hard to see that these are agravitropic- the photos mostly give the impression of spaceflight plants producing very few primary or secondary roots. Otherwise the angle of growth doesn't seem very different. Planting these

seeds in the middle of the plate instead of the top might have helped- we can't see the roots that grow "up" in these photos.

Seedlings were 12 days old. We added this information to the legend.

L168 "than what we can report here" not clear to me what this means

We modified the text to state, "what can be measured here".

L176 ref 51 refers to an oxidative burst that occurs when the lights are turned out after an extended (36 hr) day, in plants previously grown under short day conditions. That's not what you're doing here- you're doing continuous light. I'd drop this ref. Can you find another ref- besides 10- that supports the notion that continuous light is a stressor? There are no obvious reasons why it should be.

We updated the references to address this point.

L184 I don't know what flower bundles are

We defined flower bundles as a collection of open and unopened floral buds.

L207 + 221 where's the T2 data and statistical justification?

We measured telomerase activity by Q-TRAP in different pools of seedlings for T2 and T3 "super-telomerase" individuals. We chose two T3 lines for further analysis (5-2-1 with a 30.8 fold increase over wildtype and 10-3-3 with 135.8 fold increase). In Supplemental Figure 4, we added a plot of Q-TRAP results for all the T3 seedling pools to show the variance in telomerase activity levels.

L225 not clear why this promoter was chosen or why the overexpressor wasn't built into wt instead of tert, please explain rationale. Also telomere length in the OE line *looks* significantly shorter than wt here. Instead of saying that this is "within the normal range" please do a statistical analysis.

We made this series of transgenic lines for the purpose of telomerase enzyme purification. We used the *tert* mutant background as a control to ensure that the tagged version of TERT was functional (able to reconstitute telomerase enzyme activity). As suggested by the reviewer, we performed a statistical analysis of the telomere lengths for all the transformants and included this information in a new panel for Fig. 4.

L237 P values don't match those in the figure, or I'm just tired...

We thank the reviewer for noting this discrepancy. We corrected the error.

L286 telomerase, but not telomeres, right? Make your model could be made a little more clear?

We revised the text to indicate we mean the telomerase enzyme.

L366 in methods, I'm guessing seeded membranes (= time of seed imbibition) were prepared on the ground. Please clarify how long the seeded plates were left in the cold (what temp?) and dark, and if ground-based controls experienced a longer stay in the cold and dark and give us the temperature and % CO₂ used.

We expanded the methods section to include these details. We note that for the flight controls, continuous data on temperature, CO₂, and relative humidity were sent from ISS and emulated in the ground control environmental chamber on a 48-h delay.

In Conclusions, please provide some more specific conclusions- specific things that we learned from this work.

We revised the conclusion section to address this point.

Reviewer #3 (Remarks to the Author):

The authors addressed all of my comments and suggestions. Furthermore, they added new data showing that oxidative damage induced by light stress or a radiomimetic drug led to the upregulation of telomerase. They also found that plants overexpressing telomerase had lower levels of oxidative damage, as assessed by the accumulation of 8-oxoG. These additional data provide further validation for the observations made using the unique material generated during space flight. While this study does not provide deeper mechanistic insights into how telomerase implements its protective function against oxidative damage, it still presents interesting observations on how space flight affects plant physiology.

I have a few remarks:

Line 162: In the reference to single-cell transcriptomics, the authors cite papers 43 and 44. However, these studies do not contain any single-cell transcriptomics data. The authors mention in parentheses that data were also analyzed from reference 45, but no further details are provided. This should be clarified.

We rewrote the sentence to make it more clear.

Line 171: Post-transcriptional gene regulation usually refers to regulatory mechanisms between transcription and translation. However, the data in the paper do not exclude regulation at the level of translation, complex assembly, or post-translational modification.

We agree with the reviewer. Since we have no information on the mechanism, we omitted this speculation and simply stated, "The mechanism of telomerase regulation is unknown".

Line 179: Units after numerical values should not be written in parentheses.

We made the correction.

Figure 4 and corresponding text in the manuscript: The observation that overexpression of TERT from the 35S promoter does not fully rescue telomere length has already been made in Watson et al. (Genetics, 2021). These results should be discussed in the study.

We include the Watson et al. paper as corroborating evidence for our results with genetic complementation of *tert* mutants, and as evidence to address some of the concerns raised by Reviewer 2 about the 35S promoter. We discuss that while telomere length is not fully restored to wildtype levels with 35S:TERT these plants are capable of significantly extending telomeres when Ku is mutated, consistent with the notion that telomeres must be in an extendable conformation for telomerase to act.